# Mirror Speculative Decoding: Breaking the Serial Barrier in LLM Inference

## Abstract

Speculative decoding accelerates LLM inference with draft lookahead, but its effectiveness is bottlenecked by autoregressive draft generation: larger drafts improve acceptance yet also increase speculation latency overhead, capping speedup. Existing approaches such as Medusa, Hydra, EAGLE partially address draft inefficiency, but ultimately trade acceptance rates for reduced draft latency, or preserve acceptance at the cost of added overheads that limit scaling.

Modern SoCs increasingly integrate heterogeneous accelerators, most commonly GPUs and NPUs with complementary throughput and efficiency characteristics, yet existing approaches are accelerator-agnostic and usually place both draft and target on the same type of device, which leaves cross-accelerator parallelism unused. We introduce Mirror Speculative Decoding (Mirror-SD), which breaks the latency–acceptance tradeoff by launching branch-complete rollouts from early-exit signals in parallel with the target's suffix and by explicitly mapping computation across heterogeneous accelerators. In this design, the draft speculates forward token continuations for target to verify, while the target speculates correction paths for the draft, creating a bidirectional speculative process. To further reduce draft speculation latency overhead while preserving acceptance semantics, we pair Mirror-SD with speculative streaming (SS) so the draft emits multiple tokens per step. This dual strategy of combining parallel heterogeneous execution and SS pushes speculative decoding closer to its ideal regime of high acceptance while reducing speculation overhead. On SpecBench with server-scale models from 14B to 66B parameters, Mirror-SD consistently delivers realistic end-to-end gains, achieving $2.8\times$–$5.8\times$ wall-time speedups across diverse tasks representing 30% average relative improvement over the strongest baseline, EAGLE3.

## 1 Introduction

Autoregressive (AR) large language models (LLMs) have achieved state-of-the-art performance across a wide spectrum of natural language processing (NLP) tasks, yet their decoding latency remains a fundamental bottleneck, particularly for real-time applications such as interactive dialogue, code generation, and on-device assistants (Brown et al., 2020; Pope et al., 2023). Speculative decoding (SD) has emerged as a promising paradigm to mitigate this limitation by coupling a lightweight *draft model* with a larger, high-fidelity *target model* (Leviathan et al., 2023; Chen et al., 2023). In the canonical two-model SD framework, the draft model generates candidate tokens which are then verified by the target model in a serial pipeline. While this approach reduces the number of target model invocations, the sequential dependency between draft and target stages limits achievable speedups. Recent works attempt to relax the serial constraints by equipping the target itself with speculative capacity. Medusa (Cai et al., 2023) equips the target with parallel decoding heads, while EAGLE (Li et al., 2024a) introduces a dedicated speculation layer. However, the same trade-off remains: larger speculative modules improve acceptance at the cost of higher draft construction latency, while smaller ones reduce overhead but lower acceptance and limit speedup. A detailed discussion of related approaches is provided in Appendix A.

The central challenge of speculative decoding lies in reconciling these competing factors: (i) enabling *parallel execution* of draft and target models to eliminate serial dependencies, (ii) *scaling the draft capacity* to achieve higher acceptance rates without incurring proportional latency overhead, and (iii) designing *bandwidth-efficient communication protocols* that allow draft and target

to exchange token-level feedback with minimal synchronization overhead. Achieving this balance reframes speculative decoding from primarily a model-level optimization toward a system-level co-design challenge, opening the path to real-time and efficient LLM inference.

Modern System on Chip (SoC) architectures increasingly feature heterogeneous compute units that combine high-throughput GPUs with specialized neural processing units (NPUs) (Jouppi et al., 2021; Intel Corporation, 2023; Advanced Micro Devices (AMD), 2023; Apple Inc., 2023a;b; Qualcomm Technologies Inc., 2023). enabling efficient partitioning of workloads across compute substrates optimized for different performance and power trade-offs. This architectural heterogeneity motivates a division-of-labor strategy for speculative decoding, wherein the draft model operates on the NPU exploiting its efficiency for approximate inference, while the target model executes on the GPU, which is better suited for high-fidelity, throughput-critical computation. Such partitioning leverages available NPU capacity and reduces contention on the GPU, thereby improving end-to-end latency in multi-accelerator deployments.

In this work, we propose a novel architecture that operationalizes this vision by partitioning speculative decoding across heterogeneous compute units, mapping draft inference onto compute-dense NPUs and target verification onto high-throughput GPUs. This design leverages underutilized accelerator capacity, overlaps execution between models, and employs token-level feedback mechanisms to maximize acceptance while minimizing draft construction latency overhead.

## 2 Speculative Decoding: Formalization and Limits

To ground our discussion, we first formalize standard autoregressive (AR) decoding and speculative decoding (SD), establishing the baseline needed to analyze the limits of SD precisely.

**Autoregressive (AR) decoding.** Let $\mathcal{V}$ denote a finite vocabulary. We write $x_{1:m} \in \mathcal{V}^m$ for the context of length $m$ and $y_{1:T} \in \mathcal{V}^T$ for the response of length $T$ to be generated. A decoder-only AR model with parameters $\theta$ defines the conditional distribution

$$p_\theta(y_{1:T} \mid x_{1:m}) = \prod_{t=1}^{T} p_\theta\left(y_t \mid x_{1:m}, y_{<t}\right), \qquad p_\theta(\cdot \mid x_{1:m}, y_{<t}) = \mathrm{Softmax}\big(W, h_t\big), \quad (1)$$

where $h_t \in \mathbb{R}^H$ is the *next-token* representation at position $m + t$, and $W \in \mathbb{R}^{|\mathcal{V}| \times H}$ is the output head mapping hidden states to vocabulary logits (Radford & Narasimhan, 2018; Vaswani et al., 2017). Scaling inference of such models often requires distributing computation across multiple devices via tensor parallelism, which partitions per-layer parameters across devices and aggregates partial results with collectives such as ALLREDUCE (Hansen-Palmus et al., 2024; Li et al., 2024e). The per-token latency is then set by the critical path combining local compute and synchronizations.

**Speculative decoding (SD).** Speculative decoding augments a *target model* $f_{target}(\cdot \mid \cdot)$ with a computationally cheaper *draft model* $f_{draft}(\cdot \mid \cdot)$ (Leviathan et al., 2023; Chen et al., 2023). At step $t$, conditioned on the verified prefix $(x, y_{<t})$, the draft proposes a $\gamma$-token window

$$\hat{y}_{t+1:t+\gamma} \sim f_{draft}(\cdot \mid y_{<t}, x), \qquad (2)$$

which the target then verifies left-to-right, producing the largest prefix on which both models agree:

$$A_t \triangleq \max\left\{ r \in \{0, \ldots, \gamma\} : \forall j \leq r,\ \hat{y}_{t+j} = \arg\max f_{target}(\cdot \mid y_{<t+j-1}, x) \right\}. \qquad (3)$$

The agreed-upon tokens are committed as $y_{t+1:t+A_t} = \hat{y}_{t+1:t+A_t}$. If the draft and target disagree before the end of the window ($A_t < \gamma$), the target emits a correction $y_{t+A_t+1}$ and decoding resumes from $(x, y_{\leq t+A_t})$. The (window-normalized) *acceptance rate* is

$$\rho(\gamma; \phi, \theta) = \frac{\mathbb{E}[A_t]}{\gamma} \in [0, 1], \qquad (4)$$

which quantifies the expected fraction of the draft's proposals that are accepted by the target for window length $\gamma$. Let $T_{\mathrm{draft}}(\gamma; \phi)$ and $T_{\mathrm{target}}(\gamma; \theta)$ denote the wall-times to produce and to verify the window in Equations (2) and (3) (the latter includes the teacher-forced roll-forward through accepted tokens). Because verification cannot begin before speculation is available, and the *next*

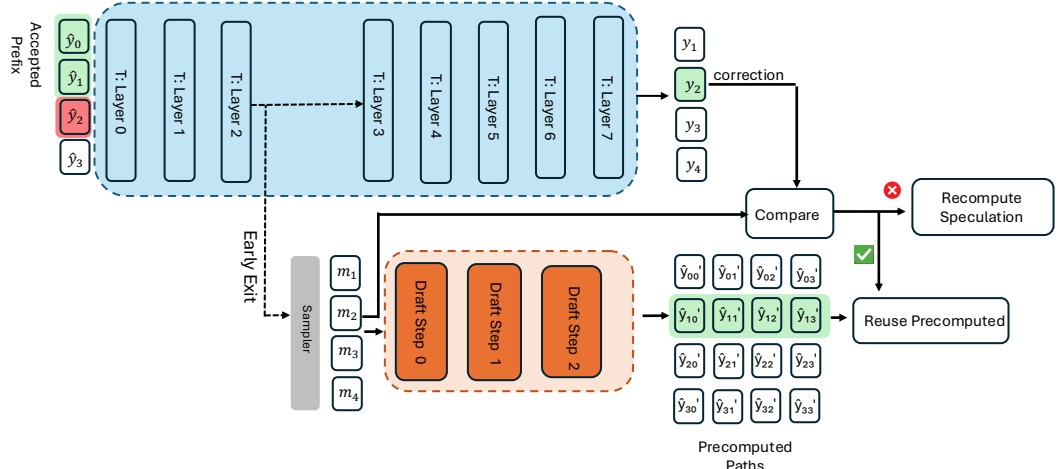

Figure 1: Mirror-SD verification and reuse (example with $\gamma = 3$, $\kappa = 1$). At early exit, the target (blue) emits $\mathcal{M}_t = \{m_1, \ldots, m_4\}$ and continues to the final layer. The draft (orange) expands $\mathcal{M}_t$ into branch-complete continuations $y'_{i0:i3}$ (grid). After verification, the target accepts $\hat{y}_0, \hat{y}_1$ and issues correction $y_2$ at depth $\tau = 2$. Reuse is possible if there exists a precomputed branch whose prefix matches the accepted tokens $(\hat{y}_0, \hat{y}_1)$ and whose node at depth $\tau$ equals $y_2$ (green). Otherwise, speculation is recomputed (See Section 3.1 for the formal rule).

speculation cannot begin before the final acceptance decision at step $t$ is known, the happen-before relation is

$$\hat{y}_{t+1:t+\gamma} \prec (\text{verification at } t) \prec \hat{y}_{t+1:t+\gamma}^{\text{next}},$$

yielding a *serial* step latency

$$T_{\text{SD}}(\gamma; \phi, \theta) = T_{\text{draft}}(\gamma; \phi) + T_{\text{target}}(\gamma; \theta). \quad (5)$$

Increasing draft capacity (larger $\gamma$, deeper/wider $f_d$) typically *increases* $\rho$ but also increases $T_{\text{draft}}$, while tiny drafts reduce $T_{\text{draft}}$ but suffer low $\rho$ (Leviathan et al., 2023; Chen et al., 2023). Equation (5) exposes the core limitation: improvements in acceptance must compensate for the added draft latency, intrinsically coupling acceptance with latency.

## 3 MIRROR SPECULATIVE DECODING

We propose *Mirror Speculative Decoding* (Mirror SD), a systems–algorithm co-design that enables parallel draft-target execution by conditioning the draft on *intermediate* target-layer distributions and reconciling via a bandwidth-light token channel. This section develops the method end-to-end—formal semantics, latency models, and a realizable tensor-parallel implementation.

### 3.1 EARLY-EXIT PROXIES AND BRANCH-COMPLETE CONCURRENT SPECULATION.

Consider a target transformer of depth $N$ with layers $L_1, \ldots, L_N$ and intermediate representations $h_t^{(\ell)}$ at step $t$. To generate high-fidelity early-exit proxies, we insert a lightweight non-linear MLP adapter $f_{\text{adapt}}(\cdot)$ that transforms the hidden state at exit layer $\ell_e < N$ before applying the shared final LM head $W_{\text{LM}}$:

$$p^{(\ell_e)}(\cdot \mid y_{<t}, x) = \text{Softmax}(W_{\text{LM}} \cdot f_{\text{adapt}}(h_t^{(\ell_e)})), \quad (6)$$

following the formulation in Pal et al. (2023a). Details of the early-exit adapter and its training procedure are provided in Appendix E.2. The resulting distribution exposes a low-bandwidth *token channel*:

$$\mathcal{M}_t = \text{Top-}\kappa(p^{(\ell_e)}(\cdot \mid y_{<t}, x)) = \{(v_i, \log \tilde{p}_i)\}_{i=1}^{\kappa}, \qquad v_i \in \mathcal{V}, \quad (7)$$

containing only the top-$\kappa$ candidate tokens and their log-probabilities. While this message is sent, the target continues its verification pass through $\mathcal{L}_{\ell_e+1}, \ldots, \mathcal{L}_N$ to form the full next-token distribution $p^{(N)}(\cdot \mid y_{<t}, x)$. Let $\gamma \in \mathbb{N}$ denote the *speculative window length*.

Given $\mathcal{M}_t$, the draft begins a *branch-complete* rollout in parallel: for each candidate $v_i$ and for every prefix length $r \leq \gamma$, it prepares a speculative continuation for the *next step* of decoding starting from $v_i$,

$$\forall i \in \{1, \ldots, \kappa\}, \ \forall r \in \{1, \ldots, \gamma\} : \qquad \hat{y}'^{(i)}_{t+1:t+r} \sim f_d(\cdot \mid y_{<t}, x, \tilde{y}_{t+1} = v_i). \tag{8}$$

While the draft's batched branches run, the target finishes verification against the currently selected draft path under the standard speculative rule and determines the first mismatch (the *correction*). Formally, let

$$A_t \triangleq \max \left\{ r \in \{0, \ldots, \gamma\} : \hat{y}_{t+j} = y^{\text{targ}}_{t+j} \ \forall j \leq r \right\}$$

be the accepted prefix length, where $y^{\text{targ}}_{t+j}$ are the target's tokens obtained from $p^{(N)}(\cdot \mid y_{<t+j-1}, x)$ (greedy/stochastic sampling). If $A_t < \gamma$, the correction occurs at index $\tau = A_t + 1$ with token

$$c_{t+\tau} \triangleq y^{\text{targ}}_{t+\tau} \sim p^{(N)}(\cdot \mid y_{<t+\tau-1}, x).$$

Let $\mathcal{T}_t$ be the hypothesis tree built at early exit from the top-$\kappa$ roots $\{v_i\}$, whose nodes at depth $r$ store the token at position $t + r$ and its precomputed continuation.

**Verification vs. reuse criterion.** At step $t$, the target accepts a prefix of length $A_t$ and issues a correction at $\tau = A_t + 1$ with token $c_{t+\tau}$. The early-exit message $\mathcal{M}_t$ induces a hypothesis tree $\mathcal{T}_t$ rooted at the top-$\kappa$ candidates, with $\text{Paths}_r(\mathcal{T}_t)$ denoting all root-to-depth-$r$ prefixes, which serve as anchors for speculative continuations. The accepted prefix is $\Pi_t = (y^{\text{targ}}_{t+1}, \ldots, y^{\text{targ}}_{t+A_t})$, and the corrected prefix extends it with the correction token, $\Pi^+_t = (\Pi_t, c_{t+\tau})$. Reuse occurs whenever this corrected prefix already appears as a path in $\mathcal{T}_t$, i.e.

$$\Pi^+_t \in \text{Paths}_\tau(\mathcal{T}_t),$$

so that only the correction must be checked while the accepted positions $1{:}A_t$ remain fixed.

*Operational selection of the next window.*

$$\hat{y}'_{t+1:t+\gamma} = \begin{cases} \text{branch rooted at } c_{t+1}, & A_t = 0 \ \wedge \ \exists i : \ v_i = c_{t+1}, \\ \text{precomputed continuation at depth } \tau \text{ along } \Pi_t, & A_t \geq 1 \ \wedge \ \Pi^+_t \in \text{Paths}_\tau(\mathcal{T}_t), \\ \text{fresh rollout from } (y_{1:t+A_t}, c_{t+\tau}), & \text{otherwise.} \end{cases}$$

In all cases, the committed output is $y^{\text{targ}}_{t+1:t+A_t}$, after which decoding advances to the next step.

**Effect of sampling width at early exit.** Let $q(\cdot) = p^{(N)}(\cdot \mid h_t)$ and $\tilde{p}(\cdot) = p^{(\ell_e)}(\cdot \mid h_t)$. We denote the top-$\kappa$ mass overlap as:

$$\Omega_\kappa = \sum_{y \in \text{Top-}\kappa(\tilde{p})} q(y). \tag{9}$$

It follows that $\mathbb{P}\big(y_{t+1} \in \text{Top-}\kappa(\tilde{p})\big) = \Omega_\kappa$, which is nondecreasing in $\kappa$ and satisfies $\lim_{\kappa \to |\mathcal{V}|} \Omega_\kappa = 1$. Larger $\kappa$ therefore reduces fallbacks requiring speculation recomputation and improves throughput, while leaving acceptance semantics intact (See Appendix B).

### 3.2 Draft execution with Speculative Streaming

For the draft model $f_d$, we employ *Speculative Streaming* (SS) (Bhendawade et al., 2024), a speculative mechanism that *verifies* previously proposed tokens while *generating* new speculative tokens *in the same forward pass* using multi-stream attention. Applying SS to the target would modify its decoding dynamics and alter the final distribution $p^{(N)}(\cdot \mid y_{<t}, x)$ (Bhendawade et al., 2024), breaking the lossless guarantee established in Appendix B. In contrast, using SS on the draft accelerates speculation generation without changing acceptance semantics, since all commitments still require verification against the unchanged target. This design leverages SS precisely where it yields additional concurrency while preserving correctness (See Appendix B). Appendix D.2 illustrates the SS mechanism and compares draft-only speedups between vanilla and SS drafts.

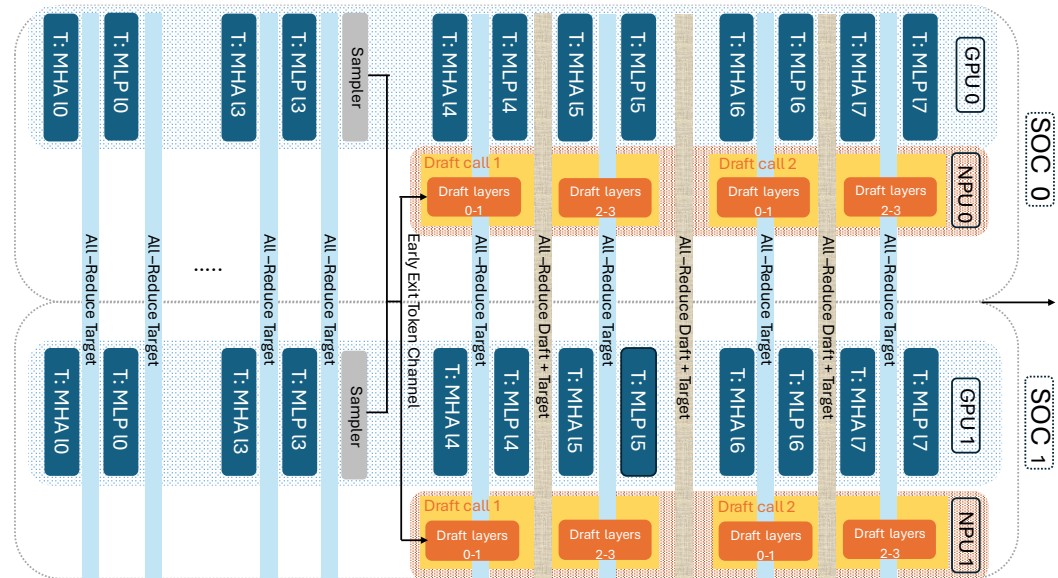

Figure 2: Heterogeneous sharding in Mirror-SD. The *target* (blue) uses Megatron-style TP with two collectives per MHA/MLP block, while the *draft* (orange) uses SPD-style sharding across $G_D$ NPUs with only two synchronizations per step. This design reduces sync cost, enlarges draft capacity, and improves acceptance without raising critical-path latency. *Note:* The beige bands labeled "All-Reduce Draft + Target" are a visual shorthand: the draft and target perform *separate* all-reduces within their own device groups, with no cross-collective coupling.

**Multi-stream attention (MSA) factorization.** Let $M_t^{(\ell)}$ denote the main-stream hidden state at layer $\ell$ and step $t$, and $S_{t,j}^{(\ell)}$ the hidden state of lookahead stream $j \in \{1, \ldots, \gamma\}$. Speculative streaming (SS) constructs attention masks so that each $S_{t,j}$ attends to the verified prefix and to lower-index lookahead streams $\{S_{t,1}, \ldots, S_{t,j}\}$, while the main stream $M_t$ attends only to the verified prefix. At the top layer, a *shared* LM head $W_{\text{LM}}^{(d)}$ projects these hidden states to token logits:

$$W_{\text{LM}}^{(d)} M_t^{(N)} \mapsto p_d(\cdot \mid h_t) \quad \text{and} \quad W_{\text{LM}}^{(d)} S_{t,j}^{(N)} \mapsto p_d(\cdot \mid h_t, j), \quad j = 1, \ldots, \gamma.$$

so a single forward pass yields both the distribution used to *verify* the prior draft and the distributions needed to *grow* the next speculative window across multiple lookahead depths. SS trains these streams with a future $n$-gram prediction objective without introducing additional heads.

**Work-conserving draft generation within Mirror-SD.** Within each Mirror-SD step, the draft must furnish a branch-complete speculative window of length $\gamma$ at the rendezvous ( Section 3.1). Under SS, a single draft *internal* step can emit $\eta_j \geq 1$ tokens by verifying the prior proposal and predicting multiple future tokens in one pass (Bhendawade et al., 2024). Consequently, the number of draft steps $J$ required to materialize $\gamma$ tokens satisfies

$$J \leq \left\lceil \frac{\gamma}{\bar{\eta}} \right\rceil, \qquad \bar{\eta} = \frac{1}{J} \sum_{j=1}^{J} \eta_j.$$

## 3.3 HETEROGENEOUS SHARDING OF MIRROR-SD

We co-schedule a depth–$N$ *target* on $G_T=8$ GPUs and a depth–$N_d$ *draft* on $G_D=8$ NPUs. The target is a pre-trained model and thus kept in its standard Megatron-style tensor parallel (TP) form (Shoeybi et al., 2019), ensuring compatibility with existing inference stacks and KV-cache layouts. In contrast, the draft is trained from scratch using the SPD architecture (Kim et al., 2025) and deployed on NPUs. We write $S$ for per–microbatch sequence length, $B$ for microbatch size, and $|\mathcal{V}|$ for vocabulary size. Figure 2 illustrates the heterogeneous sharding setup with an example configuration (target of 8 layers, draft of 4 layers); in practice, both target and draft may use different depths based on the experiment configuration.

**Target sharding** We use Megatron-style TP on the target: column-parallel $W_{qkv}$ and $W_o$ in MHA, and column/row-parallel $W_1, W_2$ in the MLP. Each transformer block performs the standard two TP collectives (attention and MLP). At early exit $\ell_e$, the target emits Top-$\kappa\big(p^{(\ell_e)}\big)$ over the token channel while continuing the verification phase; acceptance remains decided against $p^{(N)}$ and is therefore unchanged relative to vanilla SD (See Appendix B).

**Draft sharding.** The draft is trained with SPD architecture (Kim et al., 2025). We divide the $N_{\mathrm{d}}$ layers into two contiguous segments. Within each segment we instantiate $G_D$ parallel tracks; track $g \in \{1, \ldots, G_D\}$ is pinned to NPU $g$ and advances through all layers of its segment using a resident weight shard. There is no inter-NPU traffic inside a segment (See Figure 2). At the segment boundary, all tracks perform a single global synchronization to re-align tensor partitions, and a second synchronization occurs at the end of the forward pass to assemble full-width logits for the main and lookahead streams. Each internal draft step executes two all-reduce collectives on activation shards while weights remain sharded. This replaces per-layer synchronization with a fixed two-collective cost, reducing latency and enabling more parameters to be sharded across NPUs. In practice, this expands draft capacity and improves acceptance rates $\rho(\gamma; \phi, \theta)$ without increasing critical-path latency.

**Cross-accelerator rendezvous.** Mirror-SD performs two token-level exchanges per step: early-exit ($\ell_e$) and final verification ($N$). These exchanges carry $O(B\kappa)$ small items (IDs and log-probabilities) and are negligible in practice (microseconds) compared to millisecond-scale target/draft compute; they are accounted for by $T_{\mathrm{rv}}$ in the latency model.

### 3.4 LATENCY ANALYSIS

Let the target early-exit at layer $\ell_e$ in a depth-$N$ stack with per–layer times $c_\ell$, and write

$$T_{\mathrm{target}}^{1:\ell_e} = \sum_{\ell=1}^{\ell_e} c_\ell, \qquad T_{\mathrm{target}}^{\ell_e+1:N} = \sum_{\ell=\ell_e+1}^{N} c_\ell.$$

Let $\gamma$ be the speculative window length and let $T_{\mathrm{draft}}^{\mathrm{gen}}(\gamma)$ denote the time to produce a branch-complete draft window (absorbing any multi-token SS steps). We account for the two rendezvous overheads at early exit and final verification,

$$T_{\mathrm{rv}}^{(ee)}, \qquad T_{\mathrm{rv}}^{(fv)}, \qquad T_{\mathrm{rv}} \triangleq T_{\mathrm{rv}}^{(ee)} + T_{\mathrm{rv}}^{(fv)},$$

where the GPU↔NPU token exchanges carry only $O(B\kappa)$ IDs/log-probabilities.

A single Mirror-SD step consists of (i) target prefix, (ii) early-exit rendezvous, (iii) a parallel region where the target suffix overlaps the draft generation, and (iv) final rendezvous. The step latency is

$$T_{\mathrm{Mirror}} = T_{\mathrm{target}}^{1:\ell_e} + T_{\mathrm{rv}}^{(ee)} + \max\big\{T_{\mathrm{target}}^{\ell_e+1:N}, T_{\mathrm{draft}}^{\mathrm{gen}}(\gamma)\big\} + T_{\mathrm{rv}}^{(fv)}. \tag{10}$$

Let the *overlap budget* be $\Delta \triangleq T_{\mathrm{target}}^{\ell_e+1:N}$. If $T_{\mathrm{draft}}^{\mathrm{gen}}(\gamma) \leq \Delta$, the entire draft generation is hidden under the target suffix and

$$T_{\mathrm{Mirror}} = T_{\mathrm{target}} + T_{\mathrm{rv}}.$$

Otherwise the draft dominates the parallel region and

$$T_{\mathrm{Mirror}} = T_{\mathrm{target}}^{1:\ell_e} + T_{\mathrm{rv}}^{(ee)} + T_{\mathrm{draft}}^{\mathrm{gen}}(\gamma) + T_{\mathrm{rv}}^{(fv)}.$$

Thus, scaling the draft that only increases overlapped $T_{\mathrm{draft}}^{\mathrm{gen}}(\gamma)$ is *free* up to budget $\Delta$, while the token-channel transfers remain a small $O(B\kappa)$ term. We provide a full accounting of sampling/transfer costs, multi-step SS, and synchronization in Appendix C.

## 4 EXPERIMENTS

We evaluate Mirror-SD on a broad suite of generation workloads under realistic serving constraints, using server-scale decoder-only LLMs that are routinely deployed in production inference stacks across mid to large capacities, and we compare against strong speculative-decoding baselines.

## 4.1 Evaluation protocol

**Datasets and tasks.** We integrate our approach with the open-source SpecBench framework (Xia et al., 2024b) to ensure a fair, reproducible comparison against prior methods. SpecBench provides standardized prompts and pre/post-processing, sampling settings and released configs and seeds (Xia et al., 2024b). We report results on multi-turn interactive conversation (MT Bench), translation, summarization, mathematical reasoning, machine translation and retrieval-augmented generation (RAG). Context and generation lengths follow the SpecBench protocol (Xia et al., 2024b).

**Models and baselines.** We evaluate Mirror-SD on server-scale targets that are deployable in production inference stacks: Qwen3-14B and Qwen3-32B (Yang et al., 2025), Mistral-24B (Mistral, 2025), and OPT-66B (Zhang et al., 2022). For Qwen targets, we train a 0.6B-parameter draft with 2 segments and 8 tracks and deploy it on 8 NPUs as described in Section 3.3. For Mistral we train a 0.5B draft, and for OPT we train a 200M draft, both sharded as in Section 3.3 to optimize synchronization cost. All draft models are trained with SS objective described in (Bhendawade et al., 2024) on UltraChat (Ding et al., 2023). Across all target models, drafts are launched from the mid-layer early exit ($\frac{1}{2}$ of total depth) with top-$\kappa$=8 under batch size 1. Please refer to Appendix E for the effects of early-exit depth and $\kappa$. Baselines include vanilla SD, Medusa (Cai et al., 2024), Hydra (Ankner et al., 2024b), EAGLE 2/3 (Li et al., 2024b; 2025a), Recycling (Luo et al., 2024), PLD (Saxena, 2023a), SpS (Joao Gante, 2023), REST (He et al., 2024), and Lookahead (Fu et al., 2023). All baselines have public implementations in SpecBench (Xia et al., 2024b), and we use the corresponding implementations.

**Metrics.** We focus solely on efficiency, without reporting accuracy metrics, since Mirror-SD is lossless and guarantees identical outputs to the target model under the same decoding process (see Appendix B). Our two key metrics are: (i) end-to-end wall-time speedup over target-only autoregressive decoding, reported as a speedup factor; and (ii) *acceptance length*, the expected number of tokens accepted per speculative window, averaged across steps and prompts. We report greedy decoding with temperature $\tau = 0$ and stochastic decoding with $\tau = 1$. The same decoding hyperparameters are used for all methods.

**Serving configuration and reproducibility.** Target models are distributed across eight M2 Ultra GPUs using Megatron-style tensor parallelism (Section 3.3), while the draft runs on eight NPUs (Apple Inc. (2023a)). All evaluations use a fixed batch size of 1 and speculative window length $\gamma$=7; please refer to Appendix D.1 for analysis of batching effects. The token channel transmits only the top-$\kappa$ token IDs and log-probabilities in bf16. For determinism, interconnects are pinned and frequency scaling is disabled. Timings include compute, collectives, and rendezvous overhead.

## 4.2 Tri-objective analysis with an MT-Bench diagnostic

Speculative decoding couples three quantities: the speculative window $\gamma$, the acceptance length $\mathbb{E}[A_t] = \gamma \rho(\gamma; \phi, \theta)$, and drafting latency added to critical path. In vanilla SD, enlarging $\gamma$ typically boosts acceptance but also increases draft construction time since drafting is serial, yielding an upward-sloping latency curve. For Mirror-SD, the step latency follows the model in Section 3.4 (Equation (10)): as long as $T_{\text{draft}}^{\text{gen}}(\gamma) \leq \Delta$ with $\Delta = T_{\text{target}}^{\ell_e+1:N}$, increasing $\gamma$ (and thus $\mathbb{E}[A_t]$) adds *no* marginal latency; once $T_{\text{draft}}^{\text{gen}}(\gamma) > \Delta$, latency grows by the excess beyond $\Delta$. Acceptance semantics remain unchanged ( Appendix B). We validate these hypotheses on MT-Bench (Bai et al., 2024) by sweeping $\gamma$, measuring $\mathbb{E}[A_t]$ and the observed draft construction overhead added to critical path, and comparing three methods that share the same target: (i) vanilla SD with autoregressive drafts from 12M to 1.7B parameters, (ii) Mirror-SD with a 0.6M draft, and (iii) Mirror-SD with a SS draft ( Section 3.2) of 0.6B. For fairness, all approaches in Figure 3a use NPU for draft placement.

**Findings.** Vanilla SD traces an ascending surface: larger drafts increase $\mathbb{E}[A_t]$ but raise step latency commensurately. Mirror-SD shifts this surface downward by overlapping draft generation on NPUs with target verification on GPUs, revealing a near-zero-slope regime wherever $T_{\text{draft}}^{\text{gen}}(\gamma) \leq \Delta$. Adding speculative streaming further reduces $T_{\text{draft}}^{\text{gen}}(\gamma)$ by requiring fewer internal draft steps $J$ to cover the same window length $\gamma$, which extends the near-zero-slope region and pushes the surface down again. Across $\gamma$, Mirror-SD and Mirror-SD+SS dominate the Pareto frontier—achieving higher $\mathbb{E}[A_t]$ at a given latency, lower latency at a given $\mathbb{E}[A_t]$, and a wider feasible range before saturating the overlap budget defined in Section 3.4.

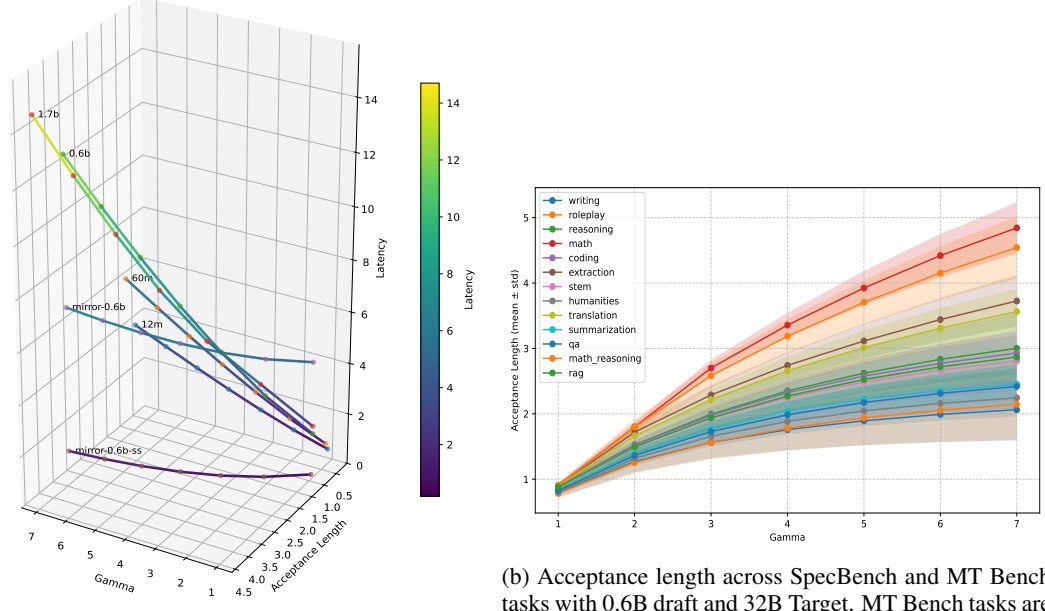

(a) Tri-objective diagnostic on MT-Bench.

(b) Acceptance length across SpecBench and MT Bench tasks with 0.6B draft and 32B Target. MT Bench tasks are reported individually.

Figure 3: (a) Speculative window $\gamma$, acceptance length, and drafting construction overhead in critical path on MT Bench. (b) Acceptance length $\mathbb{E}[A_t]$ on SpecBench and MT Bench tasks (mean $\pm$ std).

Table 1: SpecBench wall-time speedups. Mirror-SD outperforms prior methods across models, tasks, and decoding temperatures, showing consistent improvements.

| Model | Task | EAGLE3 | EAGLE2 | Hydra | Recycling | Medusa | Vanilla-SD | PLD | SpS | REST | Lookahead | Mirror-SD |
|---|---|---|---|---|---|---|---|---|---|---|---|---|
| Qwen3-14B (T=0) | Translation | 2.53x | 1.98x | 2.03x | 1.86x | 1.65x | 2.34x | 1.18x | 1.15x | 1.21x | 1.09x | **4.13x** |
| | Summarization | 2.91x | 2.19x | 2.00x | 2.30x | 1.55x | 1.76x | 2.12x | 1.87x | 1.38x | 1.30x | **3.07x** |
| | Question Answering | 3.09x | 2.39x | 2.19x | 2.13x | 1.62x | 1.81x | 1.14x | 1.31x | 1.61x | 1.27x | **3.18x** |
| | Mathematical Reasoning | 3.36x | 2.75x | 2.53x | 2.58x | 2.12x | 2.80x | 1.67x | 1.59x | 1.15x | 1.70x | **5.32x** |
| | Retrieval Aug. Generation | 2.66x | 2.13x | 2.04x | 2.06x | 1.64x | 2.02x | 1.67x | 1.75x | 1.57x | 1.32x | **3.49x** |
| | Multi-turn Conversation | 3.29x | 3.05x | 2.45x | 2.44x | 1.93x | 2.07x | 1.63x | 1.81x | 1.49x | 1.35x | **3.70x** |
| Qwen3-14B (T=1) | Translation | 1.92x | 1.81x | 1.81x | 1.78x | 1.54x | 2.19x | 1.07x | 1.04x | 1.08x | 1.03x | **3.89x** |
| | Summarization | **2.84x** | 2.05x | 1.66x | 1.84x | 1.40x | 1.50x | 1.86x | 1.40x | 1.20x | 1.13x | 2.81x |
| | Question Answering | 2.61x | 2.00x | 1.85x | 1.84x | 1.37x | 1.76x | 1.04x | 1.18x | 1.28x | 1.15x | **2.80x** |
| | Mathematical Reasoning | 3.25x | 2.54x | 2.42x | 2.29x | 2.01x | 2.53x | 1.49x | 1.42x | 1.05x | 1.39x | **5.02x** |
| | Retrieval Aug. Generation | 2.53x | 1.86x | 1.59x | 1.89x | 1.47x | 1.68x | 1.56x | 1.60x | 1.30x | 1.07x | **2.95x** |
| | Multi-turn Conversation | 3.05x | 2.78x | 2.16x | 2.15x | 1.81x | 1.98x | 1.42x | 1.41x | 1.37x | 1.24x | **3.48x** |
| Qwen3-32B (T=0) | Translation | 2.52x | 2.10x | 2.14x | 1.57x | 1.56x | 2.74x | 1.09x | 1.24x | 1.15x | 1.12x | **3.72x** |
| | Summarization | 2.98x | 2.59x | 1.98x | 1.98x | 1.56x | 2.07x | 1.82x | 1.62x | 1.38x | 1.26x | **3.14x** |
| | Question Answering | 2.76x | 2.26x | 2.17x | 1.63x | 1.81x | 2.06x | 1.17x | 1.59x | 1.70x | 1.13x | **3.04x** |
| | Mathematical Reasoning | 3.77x | 3.49x | 2.52x | 1.95x | 2.23x | 3.33x | 1.68x | 1.70x | 1.33x | 1.49x | **5.84x** |
| | Retrieval Aug. Generation | 2.65x | 2.22x | 1.92x | 1.61x | 1.59x | 2.33x | 1.42x | 1.69x | 1.76x | 1.15x | **3.42x** |
| | Multi-turn Conversation | 3.29x | 3.24x | 2.75x | 1.79x | 1.92x | 2.67x | 1.53x | 1.65x | 1.63x | 1.33x | **3.59x** |
| Qwen3-32B (T=1) | Translation | 2.36x | 1.79x | 1.90x | 1.40x | 1.42x | 2.43x | 1.03x | 1.09x | 1.03x | 1.05x | **3.15x** |
| | Summarization | 2.79x | 2.22x | 1.75x | 1.48x | 1.45x | 1.92x | 1.59x | 1.43x | 1.16x | 1.17x | **2.92x** |
| | Question Answering | 2.34x | 2.09x | 1.72x | 1.46x | 1.61x | 1.89x | 1.04x | 1.37x | 1.44x | 1.04x | **2.90x** |
| | Mathematical Reasoning | 3.45x | 3.13x | 2.35x | 1.80x | 1.66x | 2.88x | 1.36x | 1.59x | 1.20x | 1.28x | **5.08x** |
| | Retrieval Aug. Generation | 2.34x | 1.96x | 1.79x | 1.50x | 1.35x | 2.08x | 1.28x | 1.35x | 1.48x | 1.07x | **3.33x** |
| | Multi-turn Conversation | 3.14x | 2.58x | 2.29x | 1.63x | 1.73x | 2.39x | 1.34x | 1.48x | 1.47x | 1.17x | **3.28x** |

## 4.3 EFFECTIVENESS

Table 1 reports end-to-end wall-time speedups across SpecBench (Xia et al., 2024b) tasks. A clear pattern emerges: Mirror-SD shows improvements over baselines across model sizes, temperatures, and workloads. On Qwen3-14B, Mirror-SD averages $3.8\times$ acceleration with greedy sampling, compared to $2.97\times$ for the strongest prior methods; on Qwen3-32B, the average rises to $3.78\times$, eclipsing baselines at roughly $3\times$. The gains are most pronounced on long-horizon workloads (e.g., mathematical reasoning), where Mirror-SD reaches up to $5.84\times$ speedup. The improvement is driven

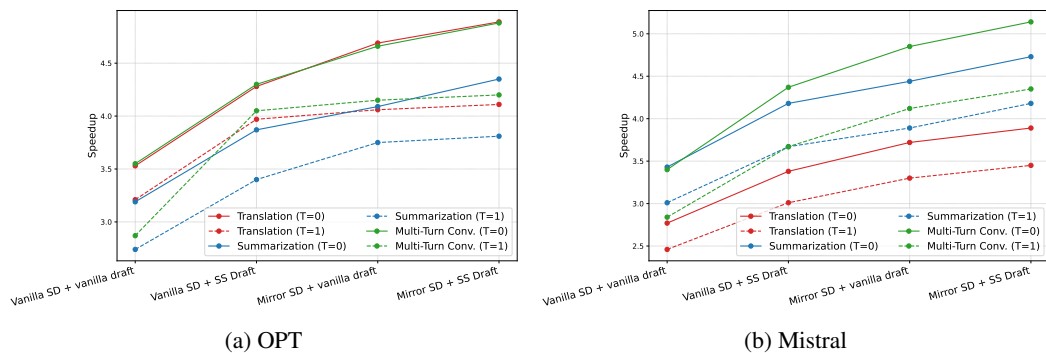

(a) OPT                                   (b) Mistral

Figure 4: Speedup for OPT and Mistral under drafting strategies across tasks and temperatures.

primarily by a larger acceptance length $\mathbb{E}[A_t]$: Mirror-SD lets us scale the draft and apply speculative streaming without paying proportional step latency, which increases the number of tokens committed per target step. Since throughput scales roughly with the expected tokens accepted per step, $S \propto 1 + \mathbb{E}[A_t]$, these acceptance gains translate directly into wall-time speedups. Retrieval-augmented generation shows a similar effect, benefitting from stable intermediate distributions that allow the draft to sustain long accepted prefixes. Even on high-entropy domains such as multi-turn conversation, where acceptance is intrinsically harder, Mirror-SD consistently delivers 3.3–3.7× acceleration compared to the 1.8-2.4× range of Hydra, Recycling or Medusa. In translation and QA, the margin is steadier but no less striking: Mirror-SD maintains a speedup edge across both greedy and stochastic decoding, validating that its improvements are insensitive to decoding regime. For an intuition grounded in the concurrency model and scaling laws behind Figure 3a, see Appendix C.

### 4.4 Generalizability across model families

To test whether the gains of Mirror-SD extend beyond Qwen, we repeat the study on two server-scale decoder-only families: Mistral-24B and OPT-66B. For each target, we hold decoding hyperparameters and draft capacity fixed and compare four variants: (1) standard speculative decoding with an autoregressive draft, (2) standard speculative decoding with a speculative-streaming draft, (3) Mirror-SD with an autoregressive draft, and (4) Mirror-SD with a speculative-streaming draft. Figure 4 reports end-to-end speedups over target-only decoding for translation, summarization, and multi-turn conversation under $\tau = 0$ and $\tau = 1$ regimes. Across both families and all tasks, the vanilla SD baseline with autoregressive-draft generation yields the smallest gains; adding speculative streaming increases throughput; switching to Mirror-SD produces a further jump; combining Mirror-SD with speculative streaming delivers the largest speedups. This progression matches the analysis in Sections 3.2 and 3.4: Mirror-SD shortens the critical path by overlapping draft generation with the target suffix, while speculative streaming reduces the draft generation time $T_{\text{draft}}^{\text{gen}}(\gamma)$ by emitting multiple tokens per internal draft step. Together, these effects allow larger acceptance lengths $E[A_t]$ without additional step latency until the overlap budget is reached, and the target's output distribution remains unchanged by construction. These results show that pairing Mirror-SD with a speculative-streaming draft generalizes across model families, delivering higher throughput without altering the base architecture or quality.

## 5 Conclusion

We introduced *Mirror Speculative Decoding* (Mirror-SD), a systems–algorithm co-design that overlaps target and draft computation, reduces draft synchronizations, and confines cross-accelerator traffic to a lightweight token channel. Deployed on heterogeneous GPU–NPU setups, Mirror-SD consistently accelerates decoding by 2.8X to 5.8X while preserving correctness. By reducing serial bottlenecks and leveraging multi-accelerator SoCs, Mirror-SD demonstrates a practical low-latency approach for large-scale LLM serving.

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

# Appendix

APPENDIX CONTENTS

## A   RELATED WORKS

**Speculative decoding with draft models.**   The original speculative decoding paradigm acceler-ates autoregressive generation by pairing a small, fast *draft* model with a larger *target* model, which verifies proposed tokens (Chen et al., 2023; Leviathan et al., 2023). This approach achieves substantial wall-time savings whenever the draft is hardware-efficient and closely aligned with the target. Domain-specialized drafts trained via distillation further improve acceptance in task-specific settings (Hong et al., 2025). Recent variants explore parallelization strategies, such as batch-axis speculation (Sun et al., 2023b) and tree-structured drafts (Miao et al., 2023; Spector & Re, 2023), to raise acceptance rates and amortize draft cost.

**Single-model approaches.**   An alternative line of work removes the explicit draft model and equips the target itself with speculative capacity. Medusa predicts multiple tokens in parallel via extra heads (Cai et al., 2023), while Hydra enforces autoregressive coupling across those heads to raise acceptance (Ankner et al., 2024b). EAGLE introduces a dedicated speculation layer (Li et al., 2024a), with EAGLE-2 enabling dynamic tree retries (Li et al., 2024b) and EAGLE-3 moving to token-level prediction with multi-layer fusion (Li et al., 2025a). Prompt-lookup decoding (PLD) and Lookahead propose suffixes by retrieval rather than generation (Saxena, 2023a; Fu et al., 2023), which is effective when prefix–continuation correlations are strong. Recycling reduces wasted work by reusing intermediate activations when speculative branches are invalidated, instead of recomputing full forwards (Luo et al., 2024). Other recent advances include structured or retrieval-based decoding policies (Yi et al., 2024a; He et al., 2024). Across the single-model designs, speculative capacity is integrated into the target stack, so larger or wider modules increase acceptance but still add work on the target's critical path; by contrast, Mirror-SD runs draft and target on heterogeneous devices and overlaps draft within the target's suffix window, converting added draft capacity into acceptance gains without inflating per-step latency proportionally.

**Dynamic and adaptive decoding.**   Beyond speculation, a range of methods accelerate inference by adapting compute during decoding. CALM (Schuster et al., 2022) and related early-exit methods reduce cost by exiting tokens at shallow layers, while skip decoding (Corro et al., 2023) mitigates key-value cache mismatch via position-dependent layer skipping. Mixture-of-Depths (MoD) (Raposo et al., 2024) routes only a subset of tokens through full blocks, yielding non-uniform FLOP allocation. Other strategies include token merging (Bolya et al., 2023) to reduce sequence length dynamically, adaptive span models (Sukhbaatar et al., 2019) that learn context windows per token, and CoLT5 (Ainslie et al., 2023) which routes tokens through heavy or light pathways. More recently, M2R2 (Bhendawade et al., 2025) introduces accelerated residual streams to improve early alignment and efficiency. Together, these approaches trade fixed per-token compute for dynamic allocation, complementing speculative decoding's strategy of parallelizing token generation.

**Positioning.**   Mirror-SD builds on these advances but takes a distinct perspective: it is a systems–algorithm co-design aimed at minimizing the *critical path* in speculative decoding. By launching drafts from intermediate target layers, overlapping draft and target compute, and confining cross-accelerator communication to lightweight token exchanges, Mirror-SD complements prior algorithmic improvements and makes speculation more effective in heterogeneous GPU–NPU deployments.

## B   CORRECTNESS: ACCEPTANCE AND DISTRIBUTION

Let $\gamma$ be the speculative window length, $N$ the number of transformer layers in the target, and let $A_t \in \{0, \ldots, \gamma\}$ denote the accepted-prefix length at step $t$. Recall that the target's final next-token distribution is $p^{(N)}(\cdot \mid h.)$ and that verification commits the longest prefix of the draft that matches the target's tokens.

**Acceptance operator (rule-level equivalence).**   For any realized draft proposal $\hat{y}_{t+1:t+\gamma}$ and realized target tokens $y_{t+1:t+\gamma}^{\mathrm{target}}$ (obtained by rolling the target with teacher forcing along the agreed prefix and stopping at the first mismatch), both vanilla SD and Mirror-SD compute

$$A_t \ = \ \max \left\{ r \leq \gamma : \ \hat{y}_{t+j} = y_{t+j}^{\mathrm{target}} \ \ \forall j \leq r \right\}. \tag{11}$$

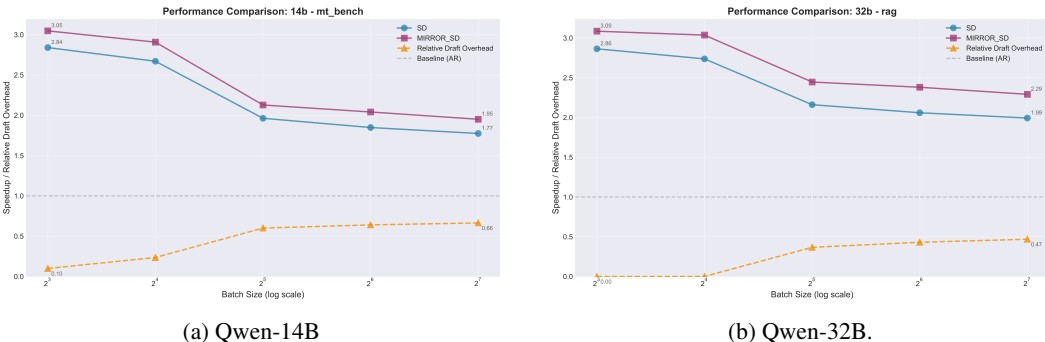

(a) Qwen-14B

(b) Qwen-32B.

Figure 5: Batching effects on speedup across tasks and scales. Both vanilla SD and Mirror-SD slow down as batch size $B$ increases due to growing draft compute and verification cost, but Mirror-SD consistently outperforms vanilla SD by preserving non-zero overlap under batching.

Equation (11) is the *same* acceptance operator in both algorithms: Mirror-SD never commits a token that was not verified against $p^{(N)}$, and any commit is exactly the longest verified prefix. Thus, Mirror-SD changes only the *schedule* by which draft proposals are produced (overlapping with target compute), not the acceptance rule.

**Distributional equivalence (when the verified draft path is identically distributed).** Fix the models $(f_{draft}, f_{target})$ and window $\gamma$. Let $\mathcal{C}_t$ be the decoding context at step $t$ (prompt and previously committed tokens), and let $\zeta_{draft}, \zeta_{target}$ collect all random seeds for draft and target sampling. Define the function

$$\mathcal{S}(\hat{y}_{t+1:t+\gamma}, y_{t+1:t+\gamma}^{\text{target}}) \;=\; \max\{r \le \gamma : \; \hat{y}_{t+j} = y_{t+j}^{\text{target}} \; \forall j \le r\},$$

so that $A_t = \mathcal{S}(\hat{y}, y^{\text{target}})$ in both procedures.

Assume the draft sequence actually presented to verification in Mirror-SD, denoted $\hat{y}_{t+1:t+\gamma}^{\text{Mir}}$, has the same conditional distribution as the vanilla draft sequence $\hat{y}_{t+1:t+\gamma}^{\text{Van}}$ given $\mathcal{C}_t$:

$$\hat{y}_{t+1:t+\gamma}^{\text{Mir}} \;\overset{d}{=}\; \hat{y}_{t+1:t+\gamma}^{\text{Van}} \;\mid\; \mathcal{C}_t. \tag{12}$$

Then, under a common coupling of $(\zeta_d, \zeta_t)$,

$$\begin{aligned}
\mathbb{P}_{\text{Mirror}}(A_t = r) &= \mathbb{P}\big(\mathcal{S}(\hat{y}^{\text{Mir}}, y^{\text{targ}}) = r\big) \\
&= \mathbb{P}\big(\mathcal{S}(\hat{y}^{\text{Van}}, y^{\text{targ}}) = r\big) \\
&= \mathbb{P}_{\text{Vanilla}}(A_t = r), \quad \forall r \in \{0, \dots, \gamma\}.
\end{aligned} \tag{13}$$

Hence the acceptance-rate statistic $\rho(\gamma; \phi, \theta) = \mathbb{E}[A_t]/\gamma$ coincides between Mirror-SD and vanilla SD.

**Sufficient condition for equation 12.** Condition equation 12 holds if the draft path used for verification in Mirror-SD is sampled from $f_{draft}(\cdot \mid h_t)$ exactly as in vanilla SD, or more generally if the branch-selection policy induces the same conditional law for the verified draft sequence as vanilla SD. Under this mild parity condition, Mirror-SD is *distributionally* identical to vanilla SD with respect to $A_t$, while still enjoying the latency benefits of overlapping draft computation with the target's suffix.

## C  LATENCY AND COMMUNICATION ANALYSIS

This appendix consolidates the latency model of Mirror-SD with its tensor-parallel (TP) communication costs.

**Draft and Target Latencies** Within one Mirror-SD step, the draft may take $J \ge 1$ *internal* steps. With speculative streaming (SS), step $j$ emits $\eta_j \ge 1$ tokens so that $\sum_{j=1}^{J} \eta_j \ge \gamma$, with average

$\bar{\eta} = \frac{1}{J} \sum_j \eta_j$ and

$$T_{\text{draft}}^{\text{gen}}(\gamma) = \sum_{j=1}^{J} (u_j^{\text{d}} + s_j^{\text{d}}), \qquad J \leq \left\lceil \frac{\gamma}{\bar{\eta}} \right\rceil.$$

Here $u_j^{\text{d}}$ is device-local compute and $s_j^{\text{d}}$ draft synchronization. For the target, each layer $\ell$ incurs $c_\ell = u_\ell^{\text{t}} + s_\ell^{\text{t}}$, giving

$$T_{\text{target}}^{1:\ell_e} = \sum_{\ell=1}^{\ell_e} c_\ell, \qquad T_{\text{target}}^{\ell_e+1:N} = \sum_{\ell=\ell_e+1}^{N} c_\ell.$$

At early exit and final verification, rendezvous costs decompose as

$$T_{\text{rv}}^{(\text{ee})} = T_{\text{samp}}^{(\text{ee})} + T_{\text{xfer}}^{(\text{ee})}, \quad T_{\text{rv}}^{(\text{fv})} = T_{\text{samp}}^{(\text{fv})} + T_{\text{xfer}}^{(\text{fv})}, \quad T_{\text{rv}} = T_{\text{rv}}^{(\text{ee})} + T_{\text{rv}}^{(\text{fv})},$$

where transfers involve only $O(B\kappa)$ IDs/log-probs and are negligible compared with compute.

**Mirror-SD Latency Law** The per-step latency is

$$T_{\text{Mirror}} = T_{\text{target}}^{1:\ell_e} + T_{\text{rv}}^{(\text{ee})} + \max\{T_{\text{target}}^{\ell_e+1:N}, T_{\text{draft}}^{\text{gen}}(\gamma)\} + T_{\text{rv}}^{(\text{fv})}. \qquad (14)$$

Let $\Delta = T_{\text{target}}^{\ell_e+1:N}$. If $T_{\text{draft}}^{\text{gen}}(\gamma) \leq \Delta$, draft work is fully hidden: $T_{\text{Mirror}} = T_{\text{target}} + T_{\text{rv}}$. Otherwise, draft cost dominates the parallel region: $T_{\text{Mirror}} = T_{\text{target}}^{1:\ell_e} + T_{\text{draft}}^{\text{gen}}(\gamma) + T_{\text{rv}}$. Compared to vanilla SD,

$$T_{\text{SD}} = T_{\text{target}}^{1:\ell_e} + T_{\text{target}}^{\ell_e+1:N} + T_{\text{draft}}^{\text{gen}}(\gamma),$$

Mirror-SD hides draft work up to $\Delta$, leaving only lightweight rendezvous terms on the critical path.

**Comparison to vanilla SD (per step).** Vanilla SD executes draft and target serially:

$$T_{\text{SD}} = T_{\text{target}}^{1:\ell_e} + T_{\text{target}}^{\ell_e+1:N} + T_{\text{draft}}^{\text{gen}}(\gamma) \; = \; T_{\text{target}} + T_{\text{draft}}^{\text{gen}}(\gamma),$$

where we write $\Delta \overset{\text{def}}{=} T_{\text{target}}^{\ell_e+1:N}$ for the *overlap budget*. Using the Mirror-SD law above,

$$T_{\text{Mirror}} = T_{\text{target}}^{1:\ell_e} + T_{\text{rv}}^{(\text{ee})} + \max\{\Delta, T_{\text{draft}}^{\text{gen}}(\gamma)\} + T_{\text{rv}}^{(\text{fv})} \; = \; T_{\text{target}} + T_{\text{rv}}, \quad \text{if } T_{\text{draft}}^{\text{gen}}(\gamma) \leq \Delta,$$

and

$$T_{\text{Mirror}} = T_{\text{target}}^{1:\ell_e} + T_{\text{draft}}^{\text{gen}}(\gamma) + T_{\text{rv}}, \quad \text{if } T_{\text{draft}}^{\text{gen}}(\gamma) > \Delta,$$

with $T_{\text{rv}} = T_{\text{rv}}^{(\text{ee})} + T_{\text{rv}}^{(\text{fv})}$.

*Per-step time saved.* The improvement is

$$\Delta T \overset{\text{def}}{=} T_{\text{SD}} - T_{\text{Mirror}} \; = \; \left( \min\{\Delta, T_{\text{draft}}^{\text{gen}}(\gamma)\} \right) - T_{\text{rv}},$$

i.e., Mirror-SD hides up to the smaller of the overlap budget and the draft time, minus lightweight rendezvous. Thus Mirror-SD is strictly faster whenever

$$T_{\text{rv}} \; < \; \min\{\Delta, T_{\text{draft}}^{\text{gen}}(\gamma)\}.$$

*Per-step speedup.* The piecewise speedup $S = T_{\text{SD}}/T_{\text{Mirror}}$ is

$$S \; = \; \begin{cases} \dfrac{T_{\text{target}} + T_{\text{draft}}^{\text{gen}}(\gamma)}{T_{\text{target}} + T_{\text{rv}}}, & \text{if } T_{\text{draft}}^{\text{gen}}(\gamma) \leq \Delta, \\[2ex] \dfrac{T_{\text{target}}^{1:\ell_e} + \Delta + T_{\text{draft}}^{\text{gen}}(\gamma)}{T_{\text{target}}^{1:\ell_e} + T_{\text{draft}}^{\text{gen}}(\gamma) + T_{\text{rv}}}, & \text{if } T_{\text{draft}}^{\text{gen}}(\gamma) > \Delta. \end{cases}$$

In practice $T_{\text{rv}}$ is $O(B\kappa)$ token/log-prob exchange and sampling, i.e., microsecond-scale, so the conditions above are typically satisfied; speculative streaming (larger $\bar{\eta}$) further reduces $J$ and $T_{\text{draft}}^{\text{gen}}(\gamma)$, making full hiding ($T_{\text{draft}}^{\text{gen}}(\gamma) \leq \Delta$) common.

**Communication Costs under TP** For $G$ devices and message size $M$ (per rank), AllReduce cost is

$$T_{\text{allreduce}}(M; G) = \alpha \log G + \beta M,$$

with $\alpha$ per-hop latency and $\beta$ per-word transfer time.

**Target:** Let $H_T$ be the target hidden width, $G_T$ its TP degree, and $S_T$ the effective tokens per collective. Each of the $N$ blocks performs two collectives on shards of size $M_T = \frac{B\,S_T\,H_T}{G_T}$, giving

$$T_{\text{target}}^{\text{comm}} = 2N \cdot T_{\text{allreduce}}\big(M_T;\, G_T\big).$$

**Draft:** Let $H_D$ be the draft hidden width, $G_D$ its TP degree, and $S_D$ the effective tokens per draft collective. Each draft *internal* step performs two collectives on shards of size $M_D = \frac{B\,S_D\,H_D}{G_D}$, so

$$T_{\text{draft-step}}^{\text{comm}} = 2\,T_{\text{allreduce}}\big(M_D;\, G_D\big), \qquad T_{\text{draft (over }J\text{ steps)}}^{\text{comm}} = 2J\,T_{\text{allreduce}}\big(M_D;\, G_D\big),$$

which is included in $T_{\text{draft}}^{\text{gen}}(\gamma)$.

**Cross-accelerator:** Token-channel exchanges remain $O(B\kappa)$ IDs/log-probs and are microsecond-scale.

## D  EXTENDED ABLATIONS & EMPIRICAL ANALYSIS

### D.1  BATCHING EFFECTS

In deployment, batching is often enabled to improve throughput and amortize GPU compute, but it is not universal: many interactive or privacy-sensitive settings prioritize per-request latency and avoid batching. To ensure completeness, we therefore also evaluate Mirror-SD under batched inference. The key question is whether speculative decoding, and Mirror-SD in particular, retains its gains when batching is enabled, or whether draft overhead grows to the point of erasing speedup. To bound the growth of draft-side computation with increasing batch size and to keep draft execution maximally hidden under the target, we *scale the draft hyperparameters with $B$*: as $B$ increases, we reduce both Top-$\kappa$ and the number of SS lookahead streams so that aggregate draft cost and the token-channel payload remain controlled. Concretely, we use $\kappa=8$ with two SS streams for $B \in \{1, 8\}$; from $B=16$ onward we use a single SS stream and progressively reduce $\kappa$: $\kappa=4$ for $B=16$, $\kappa=2$ for $B=32$, and $\kappa=1$ for $B \geq 64$.

**Observed trends.** We find that vanilla SD speedup declines steadily as batch size $B$ increases (Figure 5b). Larger batches lengthen the target verification phase both because more sequences must be processed in parallel and because batching introduces additional padding and synchronization under tensor-parallel execution. Mirror-SD also shows a downward trend with $B$, but consistently outperforms vanilla SD (Figure 5b, Figure 5a). As $B$ grows, the draft must evaluate top-$\kappa$ candidates across $\gamma$ positions for each sequence, which increases draft compute and intra-NPU communication and pushes the draft path toward a compute-bound regime. Consequently, its ability to overlap with target verification diminishes. This decreased yet positive overlap is sufficient for Mirror-SD to maintain a consistent speedup lead over vanilla SD as batching increases. In practice, batching introduces several intertwined effects: (i) the *target* takes longer, enlarging the potential overlap window; (ii) the *draft* also takes longer, and its relative overhead grows with the $\kappa \times \gamma$ expansion; (iii) autoregressive baselines slow as $B$ increases; (iv) speculative decoding slows even more, as it inherits both AR's slowdown and the draft's added work; and (v) under tensor-parallel sharding, both SD variants lose relative speedup, but Mirror-SD maintains a consistent lead by exploiting concurrency across heterogeneous accelerators.

**Relative draft overhead.** We also report a normalized "relative draft overhead" in Figure 5, defined as the fraction of draft speculation time that cannot be hidden under target verification, normalized against the total overhead of vanilla SD. This metric is dimensionless and directly reveals how much of the draft path remains exposed on the critical path. As batch size $B$ increases, the verification phase grows longer, but draft compute and intra-NPU communication grow even faster (since each sequence requires top-$\kappa$ rollouts across $\gamma$ positions). Consequently, relative draft overhead rises with $B$, aligning with the decreasing speedups observed in our batching experiments.

### D.2  DRAFT-SIDE SPEEDUPS WITH SPECULATIVE STREAMING

We quantify the internal draft gains from Speculative Streaming (SS) under the same targets and decoding settings as our main experiments. As described in Section 3.2, SS verifies previously

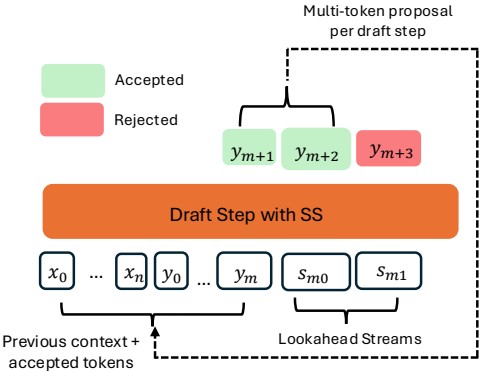

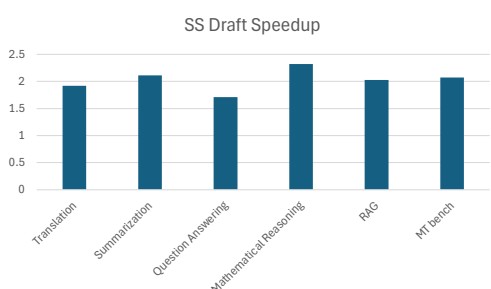

(a) Speculative Streaming (SS): each draft step proposes multiple tokens via lookahead streams; accepted tokens extend the prefix, rejected ones are dropped.

(b) Draft-only speedup from SS relative to an autoregressive draft with 3 lookahead streams. Lower $J$ for a given $\gamma$ reduces $T_{\mathrm{draft}}^{\mathrm{gen}}(\gamma)$, enlarging the overlap margin in Mirror-SD and translating into end-to-end speedups.

Figure 6: Comparison of Speculative Streaming (SS) draft dynamics (left) and resulting speedups (right).

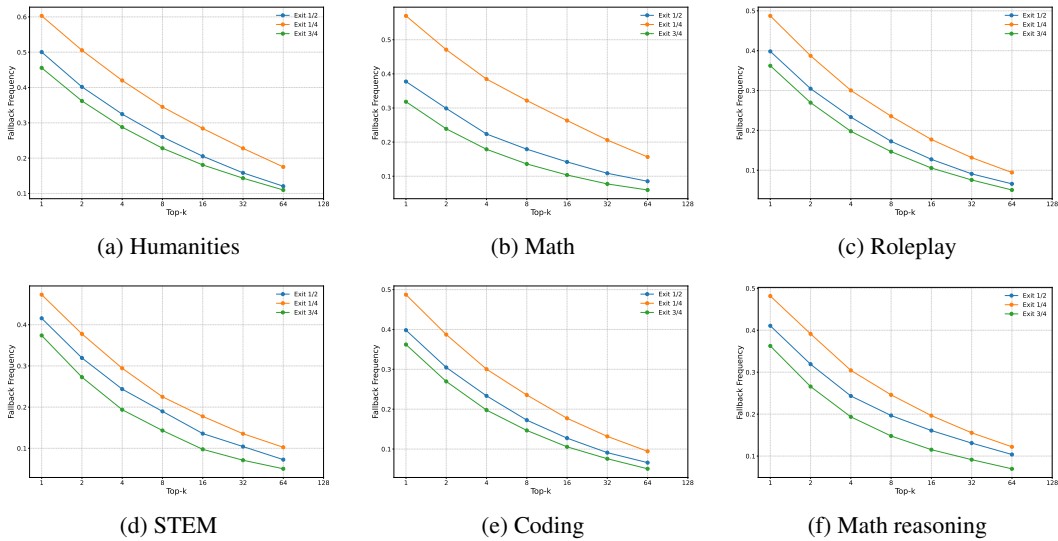

(a) Humanities      (b) Math      (c) Roleplay

(d) STEM      (e) Coding      (f) Math reasoning

Figure 7: Fallback frequency vs. Top-$\kappa$ and early-exit depth across six tasks (Humanities, Math, Roleplay, STEM, Coding, Math Reasoning). Each panel shows fallback frequency as a function of $k$ for exits at 1/4, 1/2, and 3/4 of depth; smaller values indicate fewer fallbacks and greater reuse.

proposed tokens while producing multiple new lookahead tokens in a single forward pass via multi-stream attention. Empirically, this reduces the number of draft internal steps $J$ needed to materialize a window of length $\gamma$, typically yielding $J \ll \gamma$ and a corresponding reduction in draft generation time $T_{\mathrm{draft}}^{\mathrm{gen}}(\gamma)$. Figure 6b reports the draft-only speedup of SS over a plain autoregressive draft across translation, summarization, QA, mathematical reasoning, RAG, and MT-Bench. The effect is consistent across workloads: SS achieves substantially fewer internal steps for the same $\gamma$ and, consequently, shorter $T_{\mathrm{draft}}^{\mathrm{gen}}(\gamma)$. When composed with Mirror-SD's overlap (Section 3.4), this pushes the operating point further into the zero-slope region where increases in $\gamma$ raise acceptance length $\mathbb{E}[A_t] = \gamma \, \rho(\gamma; \phi, \theta)$ without increasing step latency. Because acceptance semantics are unchanged ( Appendix B), all end-to-end gains are purely systems-level.

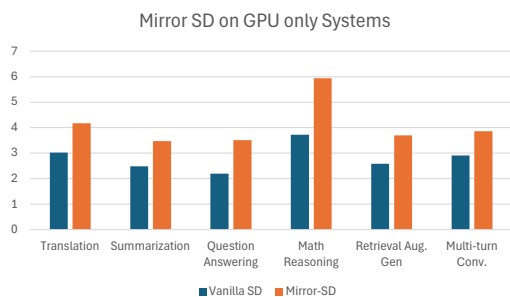

Figure 8: GPU-only evaluation of Mirror-SD at temperature $T{=}0$. The 0.6B draft runs on a single A100 GPU and the 32B target uses an 8-GPU tensor-parallel setup. All other experimental settings match Table 1, Mirror-SD consistently outperforms Vanilla-SD across all tasks.

### D.3 INFERENCE ON GPU-ONLY SYSTEMS

Mirror-SD is designed to exploit the heterogeneous accelerator topology now common in modern SoCs: a high-throughput GPU paired with a lower-power NPUs (Jouppi et al., 2021; Intel Corporation, 2023; Advanced Micro Devices (AMD), 2023; Apple Inc., 2023a;b; Qualcomm Technologies Inc., 2023). Existing speculative decoding methods do not leverage this heterogeneity; prior approaches execute both drafting and verification exclusively on GPUs, leaving substantial parallelism unused. Our primary experiments therefore target GPU–NPU systems, where Mirror-SD unlocks parallel execution of the large target model on the GPU and the lightweight draft model on the NPU with minimal communication.

For completeness, and to demonstrate hardware-agnostic applicability, we also evaluate Mirror-SD in a pure GPU setting. Here, the 0.6B draft model is executed on a single NVIDIA A100 GPU (without sharding), while the 32B target model remains sharded across the 8-GPUs via tensor-parallelism as described in Section 3.3. All early-exit heads, reuse logic, and fallback semantics remain unchanged. Although the draft model has low arithmetic intensity, draft-side latency still benefits from the substantially higher compute density and memory bandwidth of the A100 (312 TFLOPS FP16 and 1.9 TB/s HBM2e) (NVIDIA Corporation, 2020) relative to the NPU used in our main experiments (31.6 TOPS and 0.8 TB/s) (Apple Inc., 2023a). As a result, speculative rollouts are faster in both the parallel region and during fallback. Fallback frequency itself is unchanged, as it is determined solely by the target model.

As shown in Figure 8, Mirror-SD consistently improves throughput over Vanilla-SD across all six task groups. All experimental settings, model configurations, and decoding parameters match those used in Table 1. These results confirm that Mirror-SD provides reliable gains in GPU-only settings.

## E FALLBACK DYNAMICS: INFLUENCE OF TOP-$\kappa$ AND EARLY-EXIT DEPTH

### E.1 SETUP AND DEFINITIONS

At decoding step $t$, let the target's final next-token distribution be $q(\cdot) = p^{(N)}(\cdot \mid y_{<t}, x)$ and the early-exit proxy be $\tilde{p}(\cdot) = p^{(\ell_e)}(\cdot \mid y_{<t}, x)$. The target accepts a prefix of length $A_t$ and, if a mismatch occurs, issues a correction at index $\tau = A_t{+}1$ with token $c_{t+\tau}$. The draft precomputes a branch-complete window conditioned on the early-exit Top-$\kappa$ set $M_t = \{(v_i, \log \tilde{p}_i)\}_{i=1}^{\kappa}$. Reuse succeeds iff the target's correction lies on a precomputed path,

$$\Pi_t^+ \in \text{Paths}_\tau(T_t),$$

otherwise we *fallback* (re-initialize the draft from the corrected context). Let $F_t = \mathbb{1}\{\Pi_t^+ \notin \text{Paths}_\tau(T_t)\}$ and $\mathsf{FF} \equiv \mathbb{E}[F_t]$. Define the *overlap mass*

$$\Omega_\kappa(\ell_e) \stackrel{\text{def}}{=} \sum_{y \in \text{Top-}\kappa(\tilde{p})} q(y),$$

i.e., the probability under $q$ that the next token lies in the early-exit Top-$\kappa$ set.

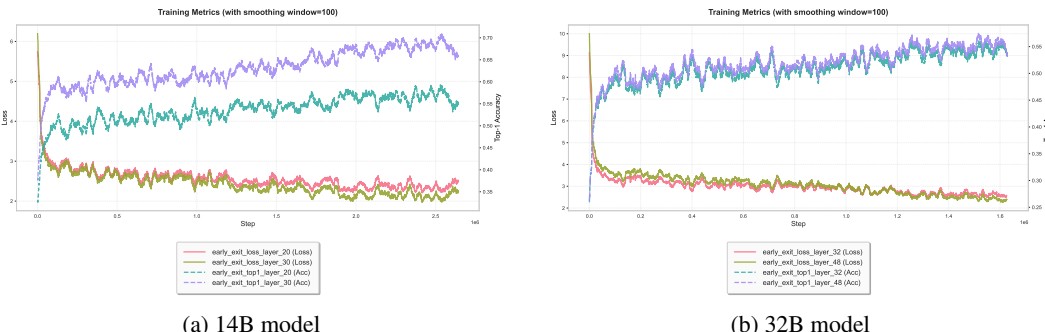

(a) 14B model              (b) 32B model

Figure 9: Early-exit training curves for the 14B and 32B target models. Each plot shows the early-exit loss and top-1 agreement for two representative exit depths. Mid-layer exits converge rapidly and achieve high agreement with the final LM head, supporting reliable branch reuse during Mirror-SD decoding.

### E.2 EARLY-EXIT TRAINING

To obtain reliable intermediate distributions for the low-bandwidth token channel defined in equation 7, we train a small set of early-exit adapters inserted at multiple depths of the target model. Specifically, we attach early-exit heads at approximately one-quarter, one-half, and three-quarters of the total transformer depth, and train all of them simultaneously. The backbone parameters remain frozen throughout training. Each early-exit head is implemented as a lightweight two-layer MLP. Given the intermediate representation $h_t^{(\ell_e)} \in \mathbb{R}^H$, the head applies a linear projection to a reduced dimension $H/2$, followed by a ReLU nonlinearity and a second linear projection back to dimension $H$. The resulting vector is then passed through the *shared* LM projection matrix $W_{\mathrm{LM}} \in \mathbb{R}^{H \times V}$, the same vocabulary projection used by the final layer of the model to produce the proxy distribution. This structure preserves the semantic geometry of the pretrained model while allowing intermediate hidden states $h_t^{(\ell_e)}$ to better align with the final-layer token distribution. The training objective is a next-token cross-entropy loss applied at each selected early-exit depth. Let $\mathcal{E} = \{\ell_1, \ell_2, \ldots, \ell_K\}$ denote the set of $K$ early-exit positions. The overall loss is

$$\mathcal{L}_{\mathrm{EE}} = \frac{1}{K} \sum_{\ell_e \in \mathcal{E}} \mathcal{L}_{\mathrm{CE}}\big(p^{(\ell_e)}(y_{t+1}), y_{t+1}\big), \tag{15}$$

where each $p^{(\ell_e)}$ is defined as in equation 6. Since the backbone remains frozen, optimization is stable and converges rapidly.

Figure 9 shows representative training curves for the 14B and 32B Qwen-3 models. Mid-layer exits typically provide strong agreement with the final LM head while maintaining low early-exit loss, enabling high-fidelity early-exit token channel. As shown in Figure 7, these intermediate distributions are accurate enough that fallback events remain infrequent when using $\kappa$-sized candidate sets.

The early-exit adapters introduce only a very small number of trainable parameters relative to the backbone, less than 0.18% of the total parameters in the 14B model and less than 0.08% in the 32B model. This makes early-exit training a lightweight and practical approach for producing high-fidelity intermediate distributions and supporting a stable token channel in Mirror-SD.

### E.3 MONOTONICITY IN $k$

**Proposition 1 (Top-$\kappa$ reduces fallback).** For a fixed early-exit layer $\ell_e$, the fallback frequency $\mathrm{FF}(\ell_e, \kappa)$ is nonincreasing in the integer $\kappa$ and vanishes as $\kappa \to |V|$:

$$\kappa_2 \geq \kappa_1 \implies \mathrm{FF}(\ell_e, \kappa_2) \leq \mathrm{FF}(\ell_e, \kappa_1), \qquad \lim_{\kappa \to |V|} \mathrm{FF}(\ell_e, \kappa) = 0.$$

*Proof.* If $A_t = 0$ (mismatch on the first token), reuse succeeds iff $y_{t+1} \in \mathrm{Top}\text{-}\kappa(p^{(\ell_e)})$, so $\Pr[F_t = 1 \mid A_t = 0] = 1 - \Omega_\kappa(\ell_e)$. If $A_t \geq 1$, the root matches $y_{t+1}$ and reuse at depth $\tau$ requires $c_{t+\tau}$ to appear

on some branch of the hypothesis tree $T_t$ seeded by Top-$\kappa(p^{(\ell_e)})$. Increasing $\kappa$ only adds roots/paths and never removes existing ones, so $\{\Pi_t^+ \in \mathrm{Paths}_\tau(T_t)\}$ is monotone in $\kappa$. Taking expectations over $t$ yields the claim. The limit follows because $\Omega_\kappa(\ell_e) \to 1$ as $\kappa \to |V|$, at which point the hypothesis tree contains all needed paths.

A useful corollary is

$$\mathsf{FF}(\ell_e, \kappa) \ \leq \ 1 - \Omega_\kappa(\ell_e),$$

which is tight when most fallbacks occur at $\tau = 1$ (high-entropy regimes).

### E.4  MONOTONICITY IN EARLY-EXIT DEPTH

**Proposition 2 (Deeper exit reduces fallback).** Fix $\kappa$. As the early-exit layer $\ell_e$ moves deeper (toward $N$), the overlap mass

$$\Omega_\kappa(\ell_e) \ = \ \sum_{y \in \mathrm{Top}\text{-}\kappa(p^{(\ell_e)})} q(y)$$

converges to its maximal value $q(S^\star)$ with $S^\star = \mathrm{Top}\text{-}\kappa(q)$; consequently $\mathsf{FF}(\ell_e, \kappa) \leq 1 - \Omega_\kappa(\ell_e)$ decreases with depth and stabilizes at its minimum for sufficiently deep exits.

*Proof.* As the layer index $\ell$ increases, the distributions $p^{(\ell)}$ approach $q$; write $\varepsilon_\ell \overset{\text{def}}{=} \|p^{(\ell)} - q\|_\infty \to 0$. Let $S_\ell = \mathrm{Top}\text{-}\kappa(p^{(\ell)})$ and $S^\star = \mathrm{Top}\text{-}\kappa(q)$. Because $S_\ell$ maximizes $p^{(\ell)}$-mass among all size-$\kappa$ sets, and any such set $A$ satisfies $|q(A) - p^{(\ell)}(A)| \leq \kappa \varepsilon_\ell$, we have

$$\Omega_\kappa(\ell) = q(S_\ell) \ \geq \ q(S^\star) - 2\kappa \varepsilon_\ell \ \xrightarrow[\ell \uparrow N]{} \ q(S^\star).$$

If the Top-$\kappa$ boundary of $q$ has margin $\Delta_\kappa > 0$, then whenever $\varepsilon_\ell < \Delta_\kappa/2$ the Top-$\kappa$ set stabilizes ($S_\ell = S^\star$) for all deeper layers, so $\Omega_\kappa(\ell) = q(S^\star)$ thereafter. Since reuse probability is monotone in the $q$-mass captured by the seed set, the bound $\mathsf{FF}(\ell_e, \kappa) \leq 1 - \Omega_\kappa(\ell_e)$ implies a (weakly) decreasing FF with depth and eventual stabilization at its minimum.

### E.5  EMPIRICAL CONFIRMATION

Figure 7 reports fallback frequency as a function of $k$ for early exits at 1/4, 1/2, and 3/4 of depth across six tasks. Two consistent trends emerge:

- **Top-$\kappa$ effect.** Increasing $k$ monotonically lowers fallback, with diminishing returns once $\Omega_\kappa$ saturates. This matches the bound $\mathsf{FF} \leq 1 - \Omega_\kappa(\ell_e)$ and reflects a higher probability that the draft's precomputed path already contains the target's correction.

- **Early-exit effect.** Holding $k$ fixed, moving the exit deeper ($1/4 \to 1/2 \to 3/4$) lowers fallback across tasks. Deeper exits raise $\Omega_\kappa$ by improving agreement between the early-exit proxy and the final distribution, so the correction token more often lies on a precomputed branch.

### E.6  PRACTICAL RECOMMENDATION

Unless otherwise noted, across all SpecBench experiments reported in Table 1 we set the Top-$\kappa$ width to $\kappa = 8$ and fix the early exit to the middle of the network ($\ell_e = N/2$, "Exit 1/2"). In practice, this mid-depth, $k=8$ configuration works well across most setups, balancing fallback probability and the overlap budget for draft precomputation.

Choosing $k$ and $\ell_e$ trades a small token-channel payload and longer precomputation for fewer fallbacks and, consequently, longer accepted prefixes per step. In Mirror-SD, the channel payload is $O(B\kappa)$ and the precomputation runs in parallel under the target suffix; thus, within the overlap budget, increasing $k$ or moving $\ell_e$ deeper reduces fallback *without* adding step latency, directly improving end-to-end throughput via larger expected acceptance length. For bandwidth-constrained deployments, $\kappa=8$, $\ell_e=N/2$ is a robust default; when acceptance is still low, increase $\kappa$ or move the exit slightly deeper (subject to the overlap budget), and when channel or memory is tight, reduce $\kappa$ or use a slightly shallower exit.

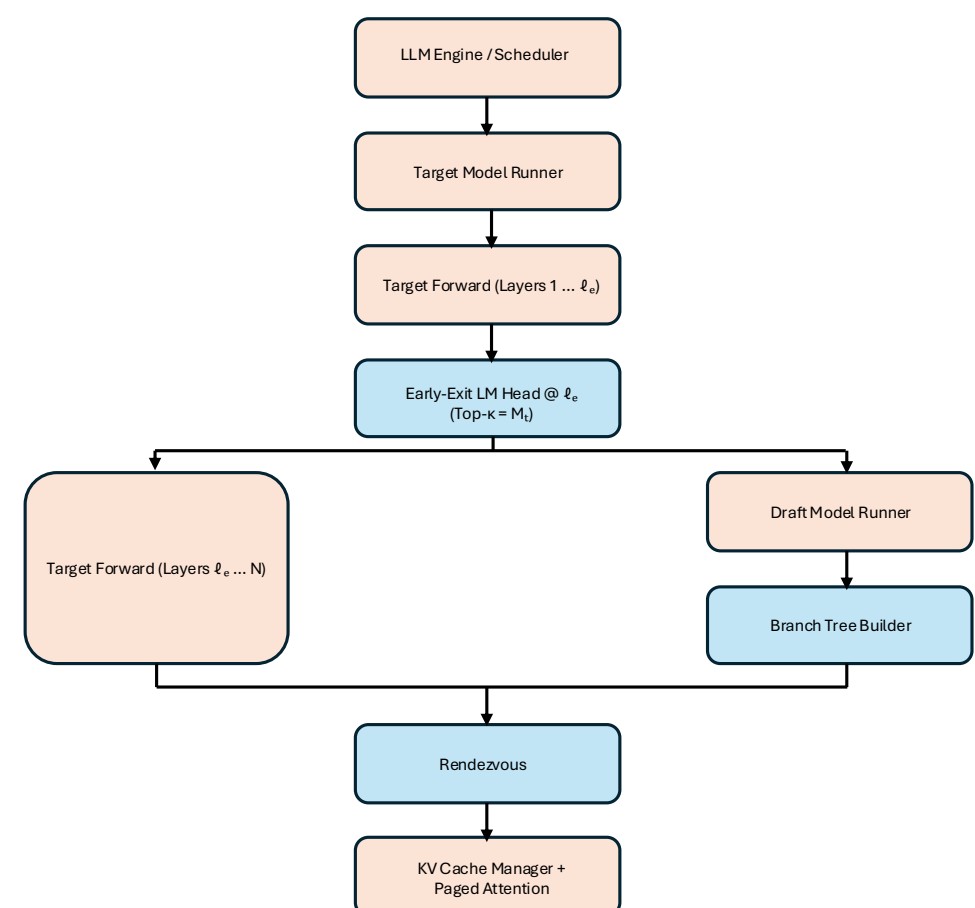

Figure 10: **Integration of Mirror-SD into vLLM.** Existing vLLM components including the scheduler, target and draft model runners, and the PagedAttention KV cache are shown in light orange. Mirror-SD adds three lightweight modules (blue): an early-exit LM head at layer $\ell_e$, a branch-tree builder for speculative rollouts, and a rendezvous module that matches the verified prefix against the speculative tree to decide on reuse. These components integrate without modifying vLLM's scheduler, memory layout, or attention kernels, preserving the single-target-forward serving invariant.

## F INTEGRATION WITH PRODUCTION INFERENCE SYSTEMS

Modern production serving stacks such as vLLM (Kwon et al., 2023b) combine continuous batching, centralized KV-cache management, and fused attention kernels to achieve high throughput. Mirror-SD integrates cleanly into this architecture without modifying the scheduler or the core batching logic. Figure 10 shows how Mirror-SD attaches to vLLM's serving stack. vLLM already provides three abstractions that are directly aligned with our design: (i) a continuous-batching scheduler that issues exactly one target forward pass per decoding tick, (ii) a split *target* and *draft* model-runner interface used by existing speculative decoders, and (iii) a block-level KV cache with prefix-sharing and branch allocation via PagedAttention (Kwon et al., 2023b). Because these components match the architectural requirements of Mirror-SD, only lightweight modules (highlighted in blue) are added, and no changes are required to scheduling, memory layout, or attention kernels.

**Early-exit instrumentation in the target runner.** As shown in the center of Figure 10, the target runner is augmented with a lightweight *early-exit head* placed after the first $\ell_e$ layers. A small MLP adapter maps $h_t^{(\ell_e)}$ into the space expected by the final LM head, after which the existing LM projection is applied and a Top-$\kappa$ operation produces the early-exit message $M_t$. Execution then bifurcates exactly as in the diagram: the target continues through layers $\ell_e{+}1{:}N$ unchanged, while

$M_t$ is forwarded to the draft runner for parallel speculation. This preserves vLLM's single-target-forward invariant and adds only a modest overhead relative to a transformer layer.

**Parallel draft execution and branch construction.** As shown on the right side of Figure 10, the second Mirror-SD component is a lightweight *branch-tree builder* that operates within vLLM's existing draft-runner abstraction. After receiving the early-exit message $M_t$, the draft model performs a branch-complete speculative rollout of depth $\gamma$, reusing the prefix KV pages provisioned by PagedAttention and allocating branch pages in exactly the same way vLLM handles divergent decoding paths. Because prefix sharing and branch-specific KV allocation are already native features of vLLM's KV manager, enabling tree-structured speculation requires no changes to the KV layout, memory management, or attention kernels.

**Verification and branch reuse.** Once the target completes layer $N$, Mirror-SD derives the accepted-prefix length $A_t$ and a correction token. The rendezvous module in Figure 10 performs a deterministic *reuse test*: if the corrected prefix matches a path in $T_t$, the corresponding chain of KV pages is reused; otherwise, the system reverts to a fresh speculative window on the next tick. This logic operates purely at the control-flow level (token IDs and page handles) and requires no changes to vLLM's scheduler, which already supports sequences advancing by different numbers of tokens per step.

**Low integration complexity.** The Mirror-SD additions shown in Figure 10 are lightweight, stateless extensions built from operations already present in vLLM, namely LM-head projections, Top-$\kappa$ extraction, KV prefix-sharing, and branch-specific page allocation. All core serving components remain unchanged: continuous batching, CUDA Graph execution, the target forward graph, and PagedAttention's KV management. As a result, Mirror-SD integrates with minimal implementation overhead while remaining fully compatible with high-throughput LLM serving in both GPU-only and heterogeneous GPU–NPU deployments.

# G ADDITIONAL EXPERIMENTAL DETAILS

## G.1 TARGET AND DRAFT SHARDING

For the experiments in Section 4, both target and draft models were distributed across *eight Apple M2 Ultra systems* (Apple Inc., 2023a), each integrating a high-throughput GPU and a dedicated Neural Engine (NPU). We allocate the target to GPUs using Megatron-style tensor parallelism and the draft to NPUs using SPD-style sharding (see Section 3.3). Each M2 Ultra consists of a dual-die package connected internally by *UltraFusion*, a die-to-die interconnect providing up to 2.5 TB/s of bandwidth while presenting the system as a single logical GPU/NPU pair (Apple Inc., 2023a). Across machines, we organize the 8 nodes into groups of 2, linked by Thunderbolt 5 interconnects (up to 120 Gbps peak bandwidth) (Apple Inc., 2024). Groups are further connected through a high-speed network fabric, providing sufficient bandwidth for inter-group synchronization with sub-millisecond latency.

In this setup, cross-accelerator token-channel communication consists only of $O(B\kappa)$ items (token IDs and a few log-probabilities), transferred via GPU→CPU→NPU copies. These messages remain negligible compared to inter-layer collectives and draft compute, consistent with the latency analysis in Section 3.4.

## G.2 DRAFT MODEL CONFIGURATION

The draft used in our experiments is a 0.6B-parameter model trained with the SPD architecture (Kim et al., 2025). It is organized into 16 transformer layers, divided into two contiguous segments of 8 layers each. Within every segment we instantiate $G_D{=}8$ parallel tracks, where track $g \in \{1, \ldots, G_D\}$ is pinned to NPU $g$ and advances through its resident shard of the segment. Each track operates with a hidden size of 256 per shard. As in Section 3.3, there is no inter-NPU traffic within a segment. Synchronization occurs only twice per forward pass: once at the segment boundary to re-align tensor partitions, and once at the output to assemble logits for both main and lookahead streams.

# H LLM USAGE STATEMENT

In preparing this manuscript, we used AI-assisted tools to check grammar and to rephrase some sentences for clarity and readability. No content, results, or analysis were generated by AI systems; all scientific contributions and conclusions are our own.

