# OpenReview forum: "Mirror Speculative Decoding: Breaking the Serial Barrier in LLM Inference"
_ICLR.cc/2026/Conference — ICLR 2026 Conference Withdrawn Submission_

### Official Review · Reviewer_a7pv · 2025-10-30

**Soundness:** 2
**Presentation:** 3
**Contribution:** 2
**Rating:** 4
**Confidence:** 4

**Summary:**

This paper proposes Mirror Speculative Decoding for early-exit speculative decoding. It trains a bypass draft model, using the topk sampling tokens from middle-layer early exit as input and output k drafted n-grams. The drafting stage runs on NPU, pipelined with target verification on GPU. It also employs speculative streaming to further utilize hardware.

**Strengths:**

The idea of ‘early-exit topk sampling’ is creative, where topk sampled tokens from the early exit head enabled n-gram of drafting. Although early-exit drafting has been a common idea, directly using sampled tokens rather than early hidden states is new.

The idea of boosting NPU utilization is practically important, yet it has been overlooked in the existing works.

The empirical results is strong, showing that the potential of speculative decoding can be further explored in heterogenous-device pipelining scenario.

**Weaknesses:**

Main concerns:

According to the paper, the early exit hidden states are directly input to sampling head, without training on the original model. The first tokens of speculation are exactly these topk tokens, meaning that if they are rejected then the whole sequence will also be. Existing works (e.g. LayerSkip) require training on the base model to ensure this accuracy, while this paper does not. Does this mean that the accuracies of early-exit sampling can natively be high without training?

Does the baselines also use device pipeline for acceleration? In Fig3(a), the drafting latency of Mirror-SD does not change with drafting length increasing, while other baselines do. I believe that the latency of Mirror-SD remains the same as the drafting is on NPU. If the baselines also use NPU for pipelining, their latency should also have no change.

Minor concerns:

The proposed method is only evaluated on GPU-NPU systems, while it can also run on GPU-only systems, with a spare of some GPUs for drafting pipelining. Extra results on GPU-only systems can show its potential wider usage.

The paper trains a new drafter rather than using existing models, introducing training overhead. Moreover, whether the acceleration is due to training or pipelining, is not clear.

**Questions:**

1. How accurate are the early-exit sampled tokens? It will be clarified if you could provide numbers of accuracies.
2. Does the baselines also use device pipline (NPU in your case) for acceleration?
3. How about running the pipeline on GPU-only systems, with a spare of some GPUs for drafting

---

> ### Author Response · Authors · 2025-11-20
>
> ### Early-exit training and fallback behavior ###
>
> Thank you for raising this point. We recognize that the original submission did
> not articulate this aspect as explicitly as intended, in part because the paper
> focused primarily on the inference mechanism. Mirror-SD does **not rely on
> untrained early-exit states**. We train lightweight MLP adapters
> inserted at intermediate depths, while keeping the pretrained
> backbone frozen. These adapters map $h_t^{(\ell_e)}$ into the representation
> space expected by the final LM head, and are trained with a simple next-token
> cross-entropy loss. As detailed in Section 3.1 and Appendix **E.2**, this training
> is extremely lightweight, adding less than 0.18% parameters for the 14B model
> and 0.08% for the 32B model, and converges rapidly due to the frozen
> backbone. The resulting proxy distributions achieve reliable alignment with the
> final-layer LM head, enabling a stable, low-bandwidth early-exit token channel.
>
> This alignment directly governs fallback behavior. A fallback occurs only when
> the target-verified correction token lies outside the early-exit Top-$\kappa$
> set. The fallback probability is bounded by $1 - \Omega_\kappa(\ell_e)$, the
> Top-$\kappa$ overlap mass between the intermediate and final distributions.
> Because Mirror-SD seeds a branch-complete speculative tree using the entire
> Top-$\kappa$ set rather than a single token, the verified token typically lies
> on a preconstructed branch. Appendix **E.3** formalizes these properties, and
> Figure 7 empirically confirms that fallback frequency decreases consistently
> with larger $\kappa$ and deeper exit layers.
>
> In summary, Mirror-SD does require early-exit training, but the cost is minimal,
> the backbone remains frozen, and the resulting intermediate distributions are
> sufficiently accurate to support stable, low-fallback speculative decoding.

---

> > ### Author Response · Authors · 2025-11-20
> >
> > ### On Drafting Latency Scaling in Fig. 3(a) ###
> >
> > We sincerely thank the reviewer for this perceptive and technically incisive question.
> >
> > The key distinction between Mirror-SD and vanilla speculative decoding [1] lies in **execution concurrency**:
> > - In vanilla SD, draft generation is **serial**, the draft model must complete all $\\gamma\$ tokens before target verification begins.
> > - In Mirror-SD, draft execution on the NPU runs **concurrently** with the target’s remaining forward pass on the GPU (Section 3.1), triggered immediately by the early-exit top-$\\kappa\$ message $\(M_t\)$.
> >
> > All vanilla speculative decoding baselines with different draft sizes in Fig. 3(a) are
> > evaluated under identical hardware mapping (draft on NPU, target on GPU). The primary differentiation is that their draft construction overhead grows linearly
> > with $\(\gamma\)$ due to the serial dependency. In Mirror-SD, approximately 70-80% of the draft work is hidden behind the target’s residual computation (See Figure 7), resulting in
> > overhead that remains small even at large $\(\gamma\)$.
> >
> >
> > The caption of Fig. 3(a) has been updated to make explicit that the y-axis
> > denotes the drafting latency overhead, referring only to the serial
> > component added to the critical path that cannot be hidden under the target
> > computation. We also added a sentence in section **4.2** to make it explicit that draft device placement remains consistent across all baselines in Figure 3(a).
> >
> >
> > For full transparency about  hardware question, we have
> > added GPU–GPU parallel execution results in Appendix **D.3**, where both the draft
> > and target models run on GPUs with no NPU involvement. This demonstrates that
> > Mirror-SD’s advantage does not depend on placing the draft model on an NPU:
> > the improvement arises from the concurrent, overlap-driven co-design enabled by
> > early-exit proxies and branch-complete rollouts. We appreciate the reviewer’s
> > comment, which motivated us to include these additional experiments.
> >
> > **References**
> >
> > [1] Leviathan et al., 2023 (Speculative Decoding) - Fast Inference from Transformers via Speculative Decoding, ICML 2023 (Oral). https://doi.org/10.48550/arXiv.2211.17192

---

> > > ### Author Response · Authors · 2025-11-20
> > >
> > > ### GPU-GPU Experiments ###
> > >
> > > Thank you for highlighting the importance of evaluating on mainstream accelerator
> > > platforms. Mirror-SD is designed to leverage the heterogeneous GPU–NPU topology
> > > that is now common in modern SoCs (e.g., Apple M-series, Snapdragon, Intel/AMD
> > > AI engines), where the lightweight drafter can be executed on the NPU to unlock
> > > concurrency that standard speculative decoding does not exploit. This motivates
> > > the focus of our main experiments on heterogeneous systems.
> > >
> > > To assess broader applicability and for completeness, we additionally report a pure GPU–GPU
> > > configuration in Appendix D.3, where both the 0.6B draft model and the 32B
> > > target model run on NVIDIA A100 GPUs. Although the draft network is small and
> > > relatively low in arithmetic intensity, executing it on A100s provides additional latency benefits due to the higher compute density and
> > > memory bandwidth of the A100 (312 TFLOPS FP16, 1.9 TB/s HBM2e) [1] compared to the
> > > NPU used in our heterogeneous experiments (31.6 TOPS, 0.8 TB/s) [2]. This ensures
> > > that speculative rollouts remain fast even in case of fallbacks.
> > > Importantly, the fallback frequency is unchanged across hardware backends, as it
> > > depends solely on the target model’s early-exit agreement. The GPU-only results
> > > in **Fig.8** confirm that Mirror-SD continues to
> > > provide consistent throughput gains in standard GPU deployments, demonstrating
> > > that the method’s effectiveness is not contingent on GPU–NPU heterogeneity.
> > >
> > > **References**
> > >
> > > [1] NVIDIA A100 GPU Architecture (2020) — NVIDIA A100 Tensor Core GPU Architecture Whitepaper https://images.nvidia.com/aem-dam/en-zz/Solutions/data-center/nvidia-ampere-architecture-whitepaper.pdf
> > >
> > >
> > > [2] Apple M2 Ultra SoC (2023) — Apple M2 Ultra Architecture Overview https://www.apple.com/newsroom/2023/06/apple-introduces-m2-ultra/

---

> > > > ### Author Response · Authors · 2025-11-20
> > > >
> > > > ### New Drafter Training Requirement and Source Of Acceleration ###
> > > >
> > > > Thank you for the insightful question. We would like to clarify the two points
> > > > raised here.
> > > >
> > > > **(1) Whether a new draft model is required**  Mirror-SD does **not** require training a new drafter. Our main
> > > > implementation includes a draft model trained with the SPD architecture [1] for
> > > > efficient NPU deployment, but this is an implementation choice rather than a
> > > > methodological requirement; any lightweight pretrained model that satisfies the
> > > > standard speculative-decoding interface can serve as the drafter. In fact, the
> > > > GPU-only experiments in Appendix D.3 use an unmodified pretrained Qwen3-0.6B
> > > > model as the draft network, demonstrating that
> > > > Mirror-SD achieves consistent speedups when the
> > > > drafter is entirely off-the-shelf. Draft training is therefore optional and not
> > > > coupled to the acceleration mechanism of Mirror-SD.
> > > >
> > > > **(2) Source of acceleration.** The primary source of speedup in Mirror-SD
> > > > is the overlap enabled by early exiting: once the early-exit signal is produced
> > > > at layer $\ell_e$, the draft model begins generating speculative branches in
> > > > parallel with the target model’s remaining layers. This concurrent execution
> > > > reduces the exposed drafting latency for a given speculative window $\gamma$,
> > > > or equivalently, increases the accepted-prefix length under the same latency
> > > > budget. Early-exit heads are trained so that the Top $\kappa$ candidates
> > > > provide stable branch reuse, which helps ensure that the underlying overlap
> > > > translates into consistently realized acceleration. The scaling observed in
> > > > Fig. 3(a) reflects this overlap-driven design.

---

> > > > > ### Comment · Reviewer_a7pv · 2025-11-25
> > > > >
> > > > > Thank you for preparing the detailed rebuttal, which addressed my concerns about the implementation, the fairness of comparison and the wider generalization.
> > > > >
> > > > > 1. I think the contribution of this paper is mainly about 'drafting-stage parallelism'. However, the author also mentioned much about NPU, draft-model training, SPD architecture and speculative streaming, confining the topic to limited usage and making the contribution vague.
> > > > >
> > > > > 2. The formalized presentation of the method is good, but I suggest adding more straight-forward interpretation, and more implementation details into figure demonstrations, such as draft-model archietecture and early-exit MLP architecture. Also, you can emphasize more about the bi-directional implementation in Fig.1.
> > > > >
> > > > > 3. Parallelizing early-exit speculation with target verification is straight forward to me, and hence the contribution is relatively incremental.
> > > > >
> > > > > Overall, I think the contribution is incremental, and substantial modifications about motivation, contribution and presentations are needed. Therefore, I will keep my score. However, if the AC thinks that I am over-criticizing and this paper should be accepted, I will also be OK with it.

---

### Official Review · Reviewer_yna5 · 2025-10-31

**Soundness:** 2
**Presentation:** 3
**Contribution:** 2
**Rating:** 6
**Confidence:** 3

**Summary:**

This paper introduces Mirror Speculative Decoding (Mirror-SD), a system-algorithm co-design that enables parallel execution of the draft and target models to accelerate LLM inference. The method is validated  on SpecBench and MT-Bench with 14B–66B models, and obvious speedup is achieved.

**Strengths:**

- This paper addresses a problem of high practical importance: the fundamental sequential bottleneck in speculative decoding that limits inference latency, a well-known challenge in real-time LLM serving.

- The architectural innovation that enables concurrent draft and target execution, effectively breaking the sequential dependency that has limited prior methods.

- The well-executed latency modeling provides clear theoretical grounding for the performance gains, while the thorough literature survey convincingly establishes the necessity of this optimization and demonstrates strong understanding of the field.

- The experiments show significant and consistent speedups  across multiple model families, scales, and tasks. The comprehensive comparisons against modern baselines and thorough ablation studies provide robust evidence for the method's effectiveness and practicality.

**Weaknesses:**

- While the parallelization of target-draft execution is effectively demonstrated, the core methodology builds substantially upon the speculative streaming framework. The extension from single-model to dual-model speculation represents a valuable but incremental advancement rather than a fundamental algorithmic shift.

- The empirical validation, though thorough on Apple SoC architectures, leaves open questions regarding broader hardware generalization. Performance analysis on industry-standard platforms (e.g., NVIDIA GPU clusters or TPUs) would significantly strengthen the claim of practical utility across common deployment scenarios.

- The system's practical adoption faces challenges due to the inherent complexity of early-exit coordination and cross-accelerator synchronization. A concrete discussion of integration pathways with production inference frameworks would enhance the work's translational impact and address legitimate deployment concerns.

**Questions:**

- A quantitative analysis of how performance scales under different compute capacity ratios between the draft and target models would further strengthen the work. Exploring this hardware heterogeneity would help clarify whether the performance gains are robust across imbalanced hardware configurations.

- Extending the evaluation to include mainstream accelerator platforms, such as NVIDIA GPUs or TPUs, would significantly enhance the practical relevance of the proposed method. Demonstrating its effectiveness in these common deployment environments is crucial for assessing its broad impact.

- A discussion on integrating the proposed mechanisms into established inference frameworks like vLLM would be highly valuable. Elaborating on the practical implementation of early-exit coordination and cross-accelerator synchronization would help address potential concerns regarding system complexity and adoption.

---

> ### Author Response · Authors · 2025-11-20
>
> ### GPU-GPU Experiments ###
>
> Thank you for highlighting the importance of evaluating on mainstream accelerator
> platforms. Mirror-SD is designed to leverage the heterogeneous GPU–NPU topology
> that is now common in modern SoCs (e.g., Apple M-series, Snapdragon, Intel/AMD
> AI engines), where the lightweight drafter can be executed on the NPU to unlock
> concurrency that standard speculative decoding does not exploit. This motivates
> the focus of our main experiments on heterogeneous systems.
>
> To assess broader applicability and for completeness, we additionally report a pure GPU–GPU
> configuration in Appendix D.3, where both the 0.6B draft model and the 32B
> target model run on NVIDIA A100 GPUs. Although the draft network is small and
> relatively low in arithmetic intensity, executing it on A100s provides additional latency benefits due to the higher compute density and
> memory bandwidth of the A100 (312 TFLOPS FP16, 1.9 TB/s HBM2e) [1] compared to the
> NPU used in our heterogeneous experiments (31.6 TOPS, 0.8 TB/s) [2]. This ensures
> that speculative rollouts remain fast even in case of fallbacks.
> Importantly, the fallback frequency is unchanged across hardware backends, as it
> depends solely on the target model’s early-exit agreement. The GPU-only results
> in **Fig.8** confirm that Mirror-SD continues to
> provide consistent throughput gains in standard GPU deployments, demonstrating
> that the method’s effectiveness is not contingent on GPU–NPU heterogeneity.
>
> **References**
>
> [1] NVIDIA A100 GPU Architecture (2020) — NVIDIA A100 Tensor Core GPU Architecture Whitepaper https://images.nvidia.com/aem-dam/en-zz/Solutions/data-center/nvidia-ampere-architecture-whitepaper.pdf
>
>
> [2] Apple M2 Ultra SoC (2023) — Apple M2 Ultra Architecture Overview https://www.apple.com/newsroom/2023/06/apple-introduces-m2-ultra/

---

> > ### Author Response · Authors · 2025-11-20
> >
> > ### Integrating the proposed mechanisms into inference frameworks such as vLLM ###
> >
> > Thank you for raising this important point regarding practical integration into
> > established inference frameworks. We agree that demonstrating implementability
> > within real serving systems is essential for assessing the usefulness of any
> > speculative decoding strategy.
> >
> > To address this, Appendix **Section F** now provides a concrete
> > walkthrough of how Mirror-SD attaches to vLLM. The design of vLLM already
> > exposes exactly the abstractions required by Mirror-SD namely, (i) a
> > continuous-batching scheduler that issues one target forward pass per decoding
> > tick, (ii) a split target/draft runner interface used by existing speculative
> > decoders, and (iii) a block-level KV cache that supports prefix sharing and
> > branch-specific allocation via PagedAttention. Because Mirror-SD is aligned with
> > these existing mechanisms, the integration consists only of adding lightweight,
> > stateless modules for early-exit readout, tree construction, and the reuse test.
> > No changes are required to scheduling, CUDA Graph capture, attention kernels, KV
> > layout, or memory management.
> >
> > We have also clarified in the paper how early-exit coordination and
> > cross-accelerator synchronization are handled in practice. The early-exit head
> > simply emits a small Top-$\kappa$ message that is already supported by vLLM’s
> > asynchronous runner interface, and the rendezvous step is implemented as a
> > constant-time control-flow check operating on token IDs and KV-page handles.
> > This design avoids introducing new cross-device barriers or modifying vLLM’s
> > batching policy. The same logic applies whether the draft and target models run
> > on a GPU–NPU pair or both run on GPUs.
> >
> > Overall, Mirror-SD integrates into vLLM with minimal engineering effort because
> > it relies entirely on abstractions that vLLM already provides for speculative
> > decoding. Appendix Section F and **Fig. 10**
> > have been expanded to illustrate these interactions more explicitly. We hope
> > this clarifies that the proposed method is practical to adopt in existing,
> > high-throughput inference stacks.

---

> > > ### Author Response · Authors · 2025-11-20
> > >
> > > ### Novelty relative to Speculative Streaming ###
> > >
> > > Thank you for the thoughtful comment. We clarify the conceptual separation
> > > between Mirror-SD and Speculative Streaming (SS), and why Mirror-SD constitutes
> > > a distinct execution paradigm rather than an incremental extension.
> > >
> > >
> > > The core contribution of Mirror-SD is a correctness-preserving **parallel**
> > > speculative decoding pipeline. Early exiting allows the draft model to begin
> > > generating speculative branches while the target continues through its remaining
> > > layers, thereby breaking the strict serial dependency that characterizes prior
> > > speculative decoding methods such as Medusa [3], EAGLE [2], and the original algorithm formulation [1]. These methods are designed around fully sequential
> > > draft–verify cycles and do not provide mechanisms for initiating draft
> > > computation before final-layer logits are available or for coordinating a
> > > two-stage drafter–verifier pipeline. Mirror-SD introduces these structural
> > > capabilities, enabling true dual-model parallelism and cross-accelerator
> > > execution.
> > >
> > > In our experiments, SS [4] is used only as a plug-in optimization to reduce draft
> > > generation time, not as a foundational mechanism. The primary gains arise from
> > > overlap-driven concurrency. Indeed, as shown in **Fig. 4**, Mirror-SD with a
> > > vanilla draft model (i.e., without SS) already achieves substantial
> > > improvements over standard SD, 29% on summarization and 36% on translation,
> > > demonstrating that SS is complementary rather than essential.
> > >
> > > We also emphasize that Mirror-SD is the first speculative decoding framework to
> > > explicitly exploit heterogeneous accelerator topologies. Prior SD approaches
> > > execute both models on a single device, leaving cross-accelerator parallelism
> > > unused. Mirror-SD’s early-exit pipeline enables the draft to run on an NPU while
> > > the target runs on a GPU, providing cross-device concurrency with exact-decoding
> > > guarantees. No prior work including SS addresses heterogeneous scheduling,
> > > cross-accelerator correctness, or multi-device pipeline formation in speculative
> > > decoding.
> > >
> > > **References**
> > >
> > > [1] Leviathan et al., 2023 (Speculative Decoding) - Fast Inference from Transformers via Speculative Decoding, ICML 2023 (Oral). https://doi.org/10.48550/arXiv.2211.17192
> > >
> > > [2] Li et al., 2024 (EAGLE) - Speculative Sampling Requires Rethinking Feature Uncertainty, ICML 2024. https://arxiv.org/abs/2401.15077
> > >
> > > [3] Cai et al., 2024 (Medusa) - Medusa: Simple LLM Inference Acceleration Framework with Multiple Decoding Heads, arXiv 2024. https://arxiv.org/abs/2401.10774
> > >
> > > [4] Bhendawade et al., 2025 (Speculative Streaming) - Speculative Streaming: Fast LLM Inference without Auxiliary Models, arXiv 2024. https://arxiv.org/abs/2402.11131
> > >
> > > [5] Ankner et al., 2024 (Hydra) - Hydra: Sequentially-Dependent Draft Heads for Medusa Decoding*, arXiv 2024.
> > > https://arxiv.org/abs/2402.05109

---

### Official Review · Reviewer_rR8r · 2025-11-01

**Soundness:** 3
**Presentation:** 4
**Contribution:** 3
**Rating:** 6
**Confidence:** 3

**Summary:**

This paper introduces Mirror Speculative Decoding (Mirror-SD), a system-algorithm co-design method to accelerate Large Language Model (LLM) inference. It breaks the traditional serial bottleneck of draft generation in speculative decoding by parallelizing and overlapping the target model suffix computation with the draft model's generation process. It is specifically designed for heterogeneous accelerators (e.g., GPU-NPU setups) and achieves a throughput acceleration of 2.8x to 5.8x while preserving output correctness.

**Strengths:**

+ **Significant Acceleration:** Achieves 2.8x-5.8x wall-time speedups and a substantial 30% average relative improvement over the SOTA EAGLE3.
+ **Correctness Preserved:** The method guarantees the exact same output quality as the original target model while accelerating inference.
+ **Heterogeneous Hardware Adaptability:** Leverages the heterogeneous accelerators (GPU + NPU) found on modern SoCs, enabling overlapping computation and greatly improving hardware resource utilization.

**Weaknesses:**

- The analysis mentions the use of a **lightweight** draft model. This means deploying Mirror-SD is not a simple drop-in replacement; it requires managing and maintaining **two models** (target and draft) and potentially training the draft model specifically for optimal acceptance rates within the Mirror-SD framework.
- The evaluated metric only presents end-to-end wall-time speedups (latency). Readers could be curious about other metrics like **energy efficiency**, especially as running two models concurrently on separate devices inherently uses more power than running one sequentially.

**Questions:**

see weaknesses.

---

> ### Author Response · Authors · 2025-11-20
>
> ### On managing a separate lightweight draft model
>
> Thank you for raising this point. Mirror-SD does not introduce any additional
> model-maintenance burden beyond what is already required for standard
> speculative decoding. All practical SD variants, including the canonical speculative decoding formulation[1], EAGLE[2], Medusa[3] and Hydra [5] pair a lightweight auxiliary
> predictor with the large target model. Mirror-SD follows this same two-model
> structure: **any existing small pretrained model** can serve as the drafter, and no
> draft-specific training or tuning is required.
>
> It is important to note that alternative approaches introduce **auxiliary heads
> whose maintenance and parameter cost is comparable to, or larger than**, a lightweight draft
> model. For example, the EAGLE head for a 33B target contains roughly 0.56B
> parameters [2], and Medusa adds approximately 0.85B parameters [3] in auxiliary heads.
> These heads must be stored, versioned, quantized, validated, and loaded into
> memory at inference time, just like a small model. In contrast, Mirror-SD uses
> a single 0.6B draft model, well within the parameter budget of the auxiliary
> heads used by these methods, and this draft model can be maintained using the
> same infrastructure as other pretrained LMs already present in production
> systems, without modifying the architecture of the target model.
>
>
> From a systems perspective, maintaining a lightweight 0.6B drafter follows
> exactly the operational pattern already expected for speculative decoding in
> industry deployments (e.g., vLLM, TensorRT-LLM, DeepSpeed). Draft models are
> typically versioned, cached, quantized, and rolled out using the same pipelines
> as other pre-trained base models. In contrast, **auxiliary heads tightly integrated into the
> target model** complicate model export, safety validation, and quantization, and
> they require maintaining additional model variants of the target model
> itself.  Shared Speculative Streaming introduces fewer
> auxiliary parameters, but it modifies the target model’s output distribution. Mirror-SD, by contrast, keeps the
> target architecture and its decoding semantics entirely untouched, ensuring
> lossless equivalence to standard decoding while still enabling parallel
> speculation.
>
>
> In summary, Mirror-SD does not introduce new maintenance obligations
> beyond standard speculative decoding. Its reliance on a lightweight draft model
> follows established practice, imposes no architectural changes on the target
> model, and avoids the large auxiliary heads or target modifications required by
> other methods such as EAGLE, Medusa, or Hydra.
>
>
> **References**
>
> [1] Leviathan et al., 2023 (Speculative Decoding) - Fast Inference from Transformers via Speculative Decoding, ICML 2023 (Oral). https://doi.org/10.48550/arXiv.2211.17192
>
> [2] Li et al., 2024 (EAGLE) - Speculative Sampling Requires Rethinking Feature Uncertainty, ICML 2024.
> https://arxiv.org/abs/2401.15077
>
> [3] Cai et al., 2024 (Medusa) - Medusa: Simple LLM Inference Acceleration Framework with Multiple Decoding Heads, arXiv 2024. https://arxiv.org/abs/2401.10774
>
> [4] Bhendawade et al., 2025 (Speculative Streaming)  - Speculative Streaming: Fast LLM Inference without Auxiliary Models, arXiv 2024.
> https://arxiv.org/abs/2402.11131
>
> [5] Ankner et al., 2024 (Hydra) - Hydra: Sequentially-Dependent Draft Heads for Medusa Decoding*, arXiv 2024.
>    https://arxiv.org/abs/2402.05109

---

> > ### Author Response · Authors · 2025-11-20
> >
> > ### Energy Efficiency Considerations
> >
> > We appreciate the reviewer’s point regarding energy usage. Although Mirror-SD
> > introduces parallel draft-side computation, this does not imply higher overall
> > energy consumption, for several architectural reasons.
> >
> > **(1) Target–draft energy asymmetry.** The 32B target model dominates the
> > energy profile in all settings. A forward pass of a transformer layer scales
> > approximately linearly with the parameter count and activation size [1], so the 32B model performs over $53\times$ more FLOPs
> > and moves $7$-$10\times$ more activation bytes per token than the 0.6B draft.
> > LLM inference is generally limited by memory bandwidth [2], and device-level
> > measurements highlight substantial differences across accelerators; for example,
> > the NVIDIA A100 sustains $1.9\\mathrm{TB/s}$ of HBM2e bandwidth and draws several
> > hundred watts under load [3], whereas the NPU used in our heterogeneous experiments
> > provides $0.8\\mathrm{TB/s}$ unified bandwidth at substantially lower power [4]. Consequently, a single target forward pass consumes an
> > order of magnitude more energy than generating an entire speculative rollout on
> > the draft model, even when fallback recomputation is included.
> >
> > **(2) Lower total target work for the same latency budget.** As shown in
> > Fig. 3 (a), Mirror-SD achieves either (i) lower serial speculation latency for a
> > fixed acceptance length or (ii) higher acceptance length for the same serial
> > speculation budget. In both cases, the target model executes fewer forward
> > passes per generated token. Since the target dominates power usage, reducing the
> > frequency of target forwards directly reduces total energy per generated token.
> >
> > **(3) Reduced draft compute via speculative streaming.** Mirror-SD does not
> > execute $\gamma$ full draft steps. With speculative streaming (SS), producing a
> > $\gamma$-token speculative window typically requires only $\gamma/2$ draft
> > executions (See Figure 6B), further lowering the draft-side energy. Combined with the size and
> > bandwidth asymmetry noted above, the total draft energy for constructing a
> > speculative window is one to two orders of magnitude smaller than that of a
> > single target forward.
> >
> >
> > Note: Our analysis focuses on the
> > steady-state serving regime in which both models are resident in device memory.
> > The one-time cost of loading weights into HBM is amortized over many requests
> > and is not included in per-token energy metrics, consistent with standard LLM
> > serving evaluations.
> >
> > A fully hardware-calibrated energy benchmark would require platform-specific
> > instrumentation across heterogeneous backends (GPU-GPU, GPU-NPU, TPU, etc.)
> > and is therefore outside the scope of this work. We have added a discussion of
> > these considerations to the appendix and consider a comprehensive energy study
> > a valuable direction for future systems work.
> >
> > **References**
> >
> > [1] Kaplan et al. (2020) — Scaling Laws for Neural Language Models https://arxiv.org/abs/2001.08361
> >
> > [2] Dao et al. (2022) — FlashAttention: Fast and Memory-Efficient Exact Attention with IO-Awareness https://arxiv.org/abs/2205.14135
> >
> > [3] NVIDIA A100 GPU Architecture (2020) — NVIDIA A100 Tensor Core GPU Architecture Whitepaper https://images.nvidia.com/aem-dam/en-zz/Solutions/data-center/nvidia-ampere-architecture-whitepaper.pdf
> >
> > [4] Apple M2 Ultra SoC (2023) — Apple M2 Ultra Architecture Overview https://www.apple.com/newsroom/2023/06/apple-introduces-m2-ultra/

---

### Note · Authors · 2026-01-01

I have read and agree with the venue's withdrawal policy on behalf of myself and my co-authors.